



# Understanding Boreal Summer UTLS Water Vapor Variations in Monsoon Regions: A Lagrangian Perspective

Hongyue Wang[1], Mijeong Park[2], Mengchu Tao[3], Cristina Peña-Ortiz[4], Nuria Pilar Plaza[5], Felix Ploeger[1,6], and Paul Konopka[1]

[1]Institute of Climate and Energy Systems, Stratosphere (ICE-4), Forschungszentrum Jülich, Jülich, Germany.
[2]U. S. National Science Foundation National Center for Atmospheric Research (NSF NCAR), Boulder, CO, USA
[3]Carbon Neutrality Research Center, Institute of Atmospheric Physics, Chinese Academy of Sciences, Beijing, China
[4]Departamento de Sistemas Físicos, Químicos y Naturales, Universidad Pablo de Olavide, 41013 Seville, Spain
[5]Centro de Investigaciones sobre Desertificación, Consejo Superior de Investigaciones Científicas (CIDE-CSIC), 46113 Moncada, Valencia, Spain
[6]Institute for Atmospheric and Environmental Research, University of Wuppertal, Wuppertal, Germany.

**Correspondence:** Paul Konopka (p.konopka@fz-juelich.de)

**Abstract.** Water vapor in the Upper Troposphere and Lower Stratosphere (UTLS) plays a crucial role in climate feedback by influencing radiation, chemistry, and atmospheric dynamics. The amount of water vapor entering the stratosphere is sensitive to cold point temperatures (CPT), making Northern Hemisphere summer monsoons more favorable for transporting water vapor into the stratosphere. This study uses a Lagrangian method to reconstruct water vapor over the Asian (ASM) and North American (NAM) monsoons, investigating their contributions to stratospheric water vapor. The Lagrangian method tracks air parcels and identifies the coldest temperature along each trajectory, contrasting with local methods that rely on vertical temperature profiles. The reconstructed water vapor fields are validated against satellite observations from SAGE III/ISS and NASA's Aura MLS. SAGE III/ISS shows stronger moisture enhancements than MLS, but both datasets reveal similar water vapor anomalies within the ASM and NAM anticyclones. Although the Lagrangian method is dry-biased compared to observations, it effectively reconstructs UTLS water vapor (correlation coefficient 0.75), capturing moist anomalies in the ASM but performing less well in the NAM. Our analysis shows that, rather than local conditions, large-scale cold point tropopause temperatures in the vicinity of the monsoons primarily drive the moisture anomalies, with NAM water vapor significantly influenced by long-range transport from South Asia. Some convection-related processes, such as east-west shifts within the ASM, are not fully captured due to unresolved temperature variability in ERA5 and missing ice microphysics. Despite biases and computational challenges, the Lagrangian method provides valuable insights into UTLS water vapor transport.

## 1 Introduction

The water vapor ($H_2O$) is a potent greenhouse gas that can amplify climate warming caused by emissions of well-mixed greenhouse gases in the atmosphere (Solomon et al., 2010; Riese et al., 2012). In the tropical Upper Troposphere and Lower Stratosphere (UTLS), water vapor is primarily controlled by the strong dehydration of moist tropospheric air entering the stratosphere at the cold point tropopause, a process commonly referred to as freeze-drying (Brewer, 1949; Randel and Park,



2019; Smith et al., 2021). However, the extent of hydration due to water vapor and ice directly injected into the stratosphere through deep, overshooting convection remains uncertain (Randel et al., 2012; Avery et al., 2017; Ueyama et al., 2020; Jensen et al., 2020; Ueyama et al., 2023; Homeyer et al., 2023; Konopka et al., 2023). Direct convection-driven transport involves rapid vertical updrafts near the convective center, occurring within minutes (Jorgensen and Lemone, 1989; Schwartz et al.,

2013). Large-scale vertical transport enhances lower stratospheric water vapor over periods of weeks to months and covers horizontal distances of thousands of kilometers (Ploeger et al., 2013). A Lagrangian approach is often more suitable for assessing atmospheric conditions near the tropopause accurately as it tracks the trajectories of air parcels. In contrast, if deep convection plays a dominant role, temperature profiles below the considered point (an instantaneous perspective) may be more relevant for calculating the amount of water vapor injected into the lower stratosphere. Lagrangian studies reconstruct water

vapor in the UTLS by identifying the coldest temperature encountered along the parcel's trajectory, often referred to as the Lagrangian CPT (Pan et al., 2018). This approach is crucial because the Lagrangian CPT reflects the long-term history of air parcels, capturing the cumulative effects of large-scale transport and temperature variability over timescales of days to even months. Several studies have successfully reconstructed UTLS water vapor using Lagrangian methods that track the minimum saturation mixing ratio of air parcels based on the Lagrangian CPT (Mote et al., 1995; Fueglistaler and Haynes, 2005; Liu

et al., 2010; Schoeberl and Dessler, 2011; Smith et al., 2021). This perspective highlights the importance of considering the history of air parcels in understanding the stratospheric water vapor distribution.

During boreal summer, enhanced UTLS water vapor mixing ratios are observed in the regions influenced by the Asian Summer Monsoon (ASM) and the North American Monsoon (NAM), both of which experience intense convection (Fu et al., 2006; Yu et al., 2020; Park et al., 2021; Clemens et al., 2022). At the same time, water vapor throughout the Northern Hemisphere

(NH) UTLS exhibits a distinct annual cycle, with the highest mixing ratios during boreal summer and autumn, and the lowest during winter, linked to the annual cycle of tropical tropopause temperatures (Randel et al., 2004; Tao et al., 2023). Several studies suggest a significant contribution of the ASM to stratospheric water vapor (e.g. Bannister et al., 2004; Wright and Gille, 2011; Rolf et al., 2018), with this contribution amounting to about 15% of the tape recorder anomaly maximum in the tropical lower stratosphere and to about 30% of the summertime maximum in the NH extratropical lowermost stratosphere

(Nützel et al., 2019). In this study, we aim to further investigate the physical processes responsible for the enhanced water vapor over the ASM and NAM regions. To achieve this, we conduct Lagrangian back-trajectory simulations utilizing the trajectory module of the Chemical Lagrangian Model of the Stratosphere (CLaMS) (McKenna et al., 2002) driven by the fifth generation European Centre for Medium-Range Weather Forecasts atmospheric reanalysis (ERA5) (Hersbach et al., 2020). We assess the performance of the Lagrangian reconstruction in capturing boreal summer UTLS water vapor distributions by

comparing simulation results with satellite datasets from SAGE III/ISS (Stratospheric Aerosol and Gas Experiment III on the International Space Station) and MLS (Aura Microwave Limb Sounder). This analysis also involves contrasting the summer monsoon regions with the entire tropics, where convection is less dominant. We utilize SAGE III/ISS for its higher vertical resolution (2 km compared to ~3 km in MLS near the UTLS region (Read et al., 2007)), which provides a more detailed representation of $H_2O$ vertical structures within the monsoon anticyclones. The MLS dataset is used for comparison due to

its extensive sampling coverage and widespread application. Furthermore, we discuss the spatial and temporal locations of the





Lagrangian cold points in relation to the observed water vapor in the monsoon regions. We also identify processes that may explain differences between the Lagrangian reconstruction and observed data, with a primary focus on deep convection, which is not fully resolved in ERA5 meteorology but can be quantified using Outgoing Longwave Radiation (OLR) derived from satellite observations (Kumar and Krishnan, 2005).

The main research questions explored in this paper are: (i) How well can stratospheric water vapor mixing ratios in the ASM and NAM observed by SAGE III/ISS and MLS be reconstructed using Lagrangian methods, especially in comparison to the tropics where such methods were successfully applied in the past (Fueglistaler et al., 2005; Hasebe and Noguchi, 2016; Smith et al., 2021)? (ii) Are the moisture anomalies observed within the ASM and NAM anticyclones locally or remotely controlled by the Lagrangian CPT and which regions are most critical? (iii) Does the Lagrangian reconstruction support the finding that

"stronger convection leads to a relatively dry stratosphere (and vice versa)" as found by Randel et al. (2015)?

This paper is organized as follows: Section 2 presents the datasets and model used, and describes the reconstruction method. Section 3 outlines our main results, including the assessment of Lagrangian water vapor reconstructions concerning both horizontal and vertical aspects, and the analysis of Lagrangian cold points. Section 4 discusses the potential causes of biases in the Lagrangian reconstruction results. Section 5 provides the conclusions.

## 70 2 Data and Method

### 2.1 Satellite observations

#### 2.1.1 MLS

The Microwave Limb Sounder (MLS) instrument on the Aura spacecraft has been providing global measurements of various atmospheric constituents since August 2004, including water vapor, ozone, carbon monoxide, sulfur dioxide, nitric acid, and ni-

trous oxide profiles using radiances from the nearest limb scan (https://www.earthdata.nasa.gov/learn/find-data/near-real-time/ mls). MLS provides a comparatively high sampling with about 3500 measurement profiles per day. Here, we use version 5.0 (v5.0) data, which provides water vapor profiles in 2.1-3.5 km vertical resolution (Lambert et al., 2017), with ~3.0 km resolution in lower stratosphere (Read et al., 2007). We focus on water vapor profiles in August from 2017 to 2019, within both Asian monsoon and North American monsoon regions. Binned data for horizontal distributions are gridded with resolution of

10°× 20°(latitude × longitude). For more details on MLS water vapor and the retrieval technique see Read et al. (2007).

#### 2.1.2 SAGE III/ISS

The Stratospheric Aerosol and Gas Experiment III on the International Space Station (SAGE III/ISS), Level 2 Solar Event Species Profiles (HDF5) Version 5.3 (v5.3) data product (https://asdc.larc.nasa.gov/project/SAGE%20III-ISS/g3bssp_53) contains comprehensive profiles of key atmospheric components collected during solar occultation events. SAGE III/ISS, launched

on February 19, 2017, employs techniques such as solar occultation, lunar occultation, and limb scattering to measure aerosols, ozone, water vapor, and other trace gases across latitudes from 70°S to 70°N. According to Davis et al. (2021), there is gener-



ally good agreement between SAGE III/ISS v5.1 and MLS v5.0 in stratospheric water vapor measurements, with SAGE III/ISS v5.1 values being approximately 0.5 ppmv (10%) drier than MLS over the 15–35 km altitude range. However, SAGE III/ISS v5.1 profiles were affected by low-quality data due to aerosol and cloud-related interferences (Park et al., 2021; Davis et al., 90 2021). These issues—such as failed retrievals and increased sensitivity to elevated aerosol loading—were largely mitigated in version 5.2 and subsequent versions, as noted in the SAGE III/ISS Data Products User's Guide (https://asdc.larc.nasa.gov/documents/sageiii-iss/guide/DPUG_G3B_v05.30.pdf).

We focus on data in August from 2017 to 2022, using v5.3 of the water vapor profiles within the entire tropics (35°S to 35°N). In addition to the years covered by MLS (2017-2019), we include three more years to enhance statistical robustness for 95 SAGE III/ISS, as the SAGE III/ISS dataset has lower horizontal and temporal sampling. The water vapor profiles are originally retrieved on a 1.0 km grid and interpolated on a 0.5 km grid from 0.5–60.0 km in altitude. In this study, we perform 1-2-1 vertically smoothing on all SAGE III/ISS water vapor profiles following Davis et al. (2021), resulting in a final vertical resolution of 2 km. The profiles are presented in units of number density. We convert the units into mixing ratio using temperature and pressure profiles from the Modern-Era Retrospective analysis for Research and Applications, Version 2 (MERRA-2). Binned 100 data used here for presenting horizontal distributions are gridded with resolution of 10°× 20°(latitude × longitude), requiring at least 5 profiles in each bin. We follow the similar procedure described in Park et al. (2021), where SAGE III/ISS v5.1 was used.

## 2.2 OLR

For representing the strength of convection, we utilize daily mean Outgoing Longwave Radiation (OLR) data from the National 105 Oceanic and Atmospheric Administration (NOAA) Climate Prediction Center (CPC) (https://psl.noaa.gov/data/gridded/data.cpc_blended_olr-2.5deg.html). The CPC blended OLR Version 1 dataset is constructed by blending level 2 OLR retrievals from NASA's Cloud and Earth Radiant Energy System broadband measurements, NOAA/NESDIS Hyperspectral measurements, and High-resolution Infrared Radiation Sounder measurements. The dataset provide daily OLR values from 1991/01/01 to the most recent available date, on a 2.5°× 2.5°(latitude × longitude) global grid. We substract the monthly average at each grid 110 point to obtain the OLR anomalies. The OLR indices used in this study are calculated by averaging the OLR anomalies within specific regions. The regions are defined as follows: (i) OLR-West: 20-30°N, 50-80°E, (ii) OLR-East: 20-30°N, 80-110°E. Here the indices are defined only for the ASM. Note that, while OLR is a commonly used proxy, it has limitations in identifying deep convection due to its reliance on infrared measurements. These measurements can misinterpret cloud-top temperatures, particularly over land and for aged anvil clouds (Liu et al., 2007).

115 ## 2.3 Models

### 2.3.1 CLaMS trajectory module

Chemical Lagrangian Model of the Stratosphere (CLaMS) is an advanced modeling framework designed for simulating the transport and chemical processes in the atmosphere (McKenna et al., 2002; Konopka et al., 2022). It employs a Lagrangian ap-





proach, where air parcels are tracked individually, allowing for a detailed and accurate representation of atmospheric dynamics
and chemistry. For this study, we use the trajectory module of CLaMS 2.0, which specifically focuses on the trajectory cal-
culations of air parcels (https://clams.icg.kfa-juelich.de/CLaMS/traj). The driving meteorological fields for these simulations
are from ERA5, with 1°× 1°(latitude × longitude) horizontal resolution, 137 vertical hybrid layers and 6-hour time interval
(Hersbach et al., 2020). We perform 180-day back-trajectory calculations for air parcels, with each air parcel launched from
the precise spatial location and time corresponding to the satellite data profiles within ASM and NAM. For the SAGE III/ISS
dataset, we set the starting points at altitudes ranging from 14.0 km to 21.0 km, with a 0.5 km interval, matching the vertical
resolution of the SAGE III/ISS profiles. For the MLS dataset, we determine the starting points by identifying the correspond-
ing geopotential height for each layer in the MLS profiles, ensuring they are set accordingly. In both cases, the trajectories are
initialized at the exact horizontal location and time of the satellite measurements, aligning the calculations closely with the
observational data.

### 2.3.2 Water vapor reconstruction

We reconstruct water vapor concentrations by identifying the CPT. From the local perspective, cold points are the lowest tem-
peratures observed along local vertical profiles. From the Lagrangian perspective, the cold points are defined as the minimum
temperatures encountered along the back-trajectories of air parcels, after interpolating the ERA5 temperature and pressure
data along the back-trajectories. The reconstructed stratospheric water vapor concentrations are calculated using the following
formulas: $H_2O_{\text{ppmv}} = 1.0 \times 10^6 \cdot e_{\text{sat}}/(P - e_{\text{sat}})$, where the saturation vapor pressure $e_{\text{sat}}$ is given by $e_{\text{sat}} = 10^{\left(\frac{A}{CPT} + B\right)}/100$.
Here, $A = -2663.5$, $B = 12.537$, $CPT$ is the temperature in K, and $P$ is the pressure in hPa (Sonntag, 1994).

In the following, we present the results of three experiments based on three types of reconstructions: LOC, LAG_single
and LAG. LOC uses the minimum temperatures along the local temperature profiles, i.e., the local CPTs, to calculate the
reconstructions. On the other hand, LAG_single and LAG both use the Lagrangian CPTs to calculate the reconstructions, but
they employ different methods for identifying the CPTs. For LAG_single, we initiate back-trajectoy simulation from every
single satellite observation point in the UTLS, using the observed altitude, longitude, and latitude of that point. For LAG,
we reconstruct each measurement point from an ensemble of trajectories by adding 50 additional starting points around each
observation point, spaced 10 meters vertically above and below the observation point, and then consider the ensembles of
back-trajectories. For example, if the observation point is at 16.0 km (as in the SAGE III/ISS dataset), we set the starting point
at 16.0 km and add 50 more points from 15.25 km to 16.25 km, with 0.01 km (10 meters) intervals. The final reconstruction
value for this observation point is calculated by averaging the reconstruction values from all 51 back-trajectories, to enhance
the vertical sampling around the original observation point. We focus on increasing the sampling in the vertical direction rather
than the horizontal because vertical wind shear in the atmosphere tends to redistribute air horizontally. Over time, air parcels
stretch into thin, wide layers, similar to pancake-like structures, as a result of quasi-isentropic flow. This natural horizontal
spreading reduces the need for more horizontal sampling, as parcels dilute gradually through stirring. Additionally, given the
vertical resolution of MLS and SAGE III/ISS data (3 km and 2 km, respectively), it is more important to increase the vertical
sampling in our trajectory calculations to better reconstruct the water vapor mixing ratios. For further details on this process,





we refer to Haynes and Anglade (1997), which explains how differential advection in the atmosphere drives vertical mixing and stretching of air parcels. All the back-trajectories are categorized into two groups: those that cross the tropopause, which

represent Troposphere-to-Stratosphere Transport (TST), and those that do not, referred to as non-TST. TST trajectories are defined as those with starting points (or observation points) located above 370 K potential temperature and traceable back to below 340 K potential temperature. For TST trajectories, the reconstructed water vapor concentrations are calculated using the Lagrangian CPTs. For non-TST trajectories, the reconstruction values are defined as the smaller values between the saturation values, calculated using Lagrangian CPTs, and the zonal monthly climatological water vapor concentrations (MLS) at the

back-trajectory endpoints (the earliest points in time).

## 3  Results

### 3.1  Performance of Lagrangian water vapor reconstruction

#### 3.1.1  Spatial distributions

In boreal summer, both the ASM and NAM regions feature high water vapor concentrations within the UTLS. Figure 1a and

1b present the horizontal distribution of water vapor in August at ~16.5 km (around 100 hPa and 380 K potential temperature) based on SAGE III/ISS and MLS satellite observations. The distributions from both satellite datasets show consistent patterns, with notably high water vapor concentrations located in the two main monsoon regions, the ASM (15°-35°N, 50°-150°E) and NAM (10°-35°N, 160°-80°W). The high values from SAGE III/ISS (exceeding 7 ppmv) are higher than the values from MLS (5-6 ppmv). Figure 1c-d show reconstructed water vapor from Experiment LAG with more than 80% TST trajectories

(Sect. 2.3.2), based on the profiles from SAGE III/ISS and MLS, respectively. The large-scale patterns in the reconstructions are consistent with the observations, but there are obvious dry biases throughout the entire tropics, especially in the NAM region.

The anomalies shown in Fig. 1e-h are derived by subtracting the average values of the entire tropics. The observed anomalies from SAGE III/ISS in Fig. 1e illustrate that during monsoon season, water vapor concentrations in the ASM and NAM increase

by 1-2 ppmv, while from MLS (Fig. 1f), the increases are slightly smaller, amounting to approximately 1 ppmv. Comparing the anomalies of both reconstruction and observation, the reconstruction captures the enhancements in water vapor concentrations and their locations, particularly in the ASM region, where the elevation of water vapor (1-2 ppmv) is nearly fully reproduced, though with slightly more limited coverage However, the reconstruction performs unsatisfactorily in the NAM region, with an increase of less than 0.5 ppmv being reproduced.

Figure 2 analyzes the vertical structure of the observed and reconstructed water vapor profiles, averaged over the three regions of interest: tropics, ASM, and NAM. As expected, all water vapor values along the profiles increase from the stratosphere to the UTLS region, and the reconstructed profiles partially reproduce the observed enhancements in both water vapor concentrations and variations. The reconstructed profiles exhibit maximum dry biases of up to 5 ppmv in the upper troposphere below 16 km. At 16.5 km, in the UTLS, the dry biases are 2-3 ppmv. Similar dry biases are reported by Liu et al. (2010), who found



that water vapor predictions for the stratospheric overworld exhibit dry biases of up to -50% ± 10%, which they attributed to missing cloud microphysics. Above 19.0 km, the biases in the reconstructions are 1-2 ppmv smaller when both TST and non-TST trajectories are considered (Fig. 2), compared to when only TST trajectories are used (Fig. S1). The cyan squares represent the percentage of TST trajectories relative to the total number of trajectories, indicating that non-TST trajectories account for more than 95% above 19.0 km. This suggests that water vapor concentrations in the higher stratosphere align more

closely with climatological values.

For monsoon regions, the main structures of both observed and reconstructed profiles are similar to those in the tropics, though there are some noticeable differences. From the observed profiles, UTLS water vapor concentrations in monsoon regions are 2-4 ppmv higher than in the entire tropics. The enhancements of water vapor in the ASM from SAGE III/ISS (Fig. 2c) are the most significant, exceeding 4 ppmv at 15.5 km. For the reconstructions, compared to the profiles in the tropics,

the reconstructed profiles in the ASM capture an increase of ~1 ppmv in the UTLS (Fig. 2c-d), and the dry biases increase to 2-5 ppmv. In the NAM, the reconstructed profiles show insignificant differences compared to those in the tropics (Fig. 2e-f), with biases of 3-4 ppmv. Comparing the profiles from SAGE III/ISS (left) with those from MLS (right), the higher vertical resolution profiles from SAGE III/ISS show more strongly enhanced water vapor concentrations and clearer peak values in the UTLS for the entire tropics, including the monsoon regions, especially in the ASM. Additionally, the reconstructions based on

SAGE III/ISS and MLS resemble each other, both capturing an enhancement of ~1 ppmv of water vapor in the ASM but not in the NAM.

The SAGE III/ISS dataset, with its higher vertical resolution compared to MLS, captures more features of water vapor variations in the UTLS, while MLS may lose information due to its coarser layers. However, the limited and uneven sampling of SAGE III/ISS might restrict its ability to reveal spatial features, which could also be the main reason for the slight differences

between the reconstructions based on the two datasets. The diagnosed dry biases in the reconstructed water vapor concentrations could be due to systematic temperature differences in the ERA5 reanalysis (Tegtmeier et al., 2020), but are probably mainly attributable to moistening processes not included in the trajectory simulations, such as deep convection and related ice injection. Nonetheless, the analysis above implies that the reconstruction using Lagrangian CPT generally reproduces the overall horizontal patterns and vertical structures across the tropics and within the ASM, but it does not adequately capture the

patterns in the NAM.

### 3.1.2   Lagrangian versus local reconstruction

To assess the performance of the reconstructions from different experiments, we present the correlation coefficients between observed and reconstructed water vapor concentrations in Fig. 3. The x-axis in each plot shows the length of the backward period used for the trajectory calculations. In addition to LAG based on both SAGE III/ISS and MLS datasets, we include

LAG_single and LOC based on SAGE III/ISS to compare local and Lagrangian methods. LOC shows the lowest correlation coefficients: -0.12 in the tropics, 0.07 in the ASM, and -0.17 in the NAM, indicating that it is not the local cold point that determines water vapor concentrations, neither in the deep tropics (as e.g. shown by Fueglistaler et al., 2005) nor in the summertime monsoon regions. Furthermore, the simulation with higher resolution (LAG) demonstrates significantly better





performance measured as higher correlation, compared to the simulation with lower resolution (LAG_single). This is because
LAG_single relies on individual trajectories, which are highly sensitive to the initial position of the air parcel and small
variations in the ERA5 wind and diabatic heating fields. As a result, the uncertainty in the parcel's position increases as the
trajectory is traced backward. On the other hand, LAG uses trajectory ensembles, which account for uncertainties in the initial
conditions and slight differences in the ERA5 fields. By averaging over these ensembles, the reconstruction becomes more
robust and accurate, as it better captures the uncertainties inherent in the system. For LAG, changing the dataset does not
significantly affect reconstruction performance across the three regions. This suggests that the results based on SAGE III/ISS
are generally representative and reliable for the objectives of this study, despite its less horizontal and temporal sampling
compared to MLS. Moreover, the comparison across the three regions shows no significant distinctions, indicating that the
efficiency of the Lagrangian reconstruction does not vary noticeably between monsoon regions and the entire tropics.

The reconstruction of water vapor using the Lagrangian method aims to find the minimum saturation mixing ratio along
the trajectory, and therefore the backward time length of the simulation might influence the results. As shown in Fig. 3,
all Lagrangian experiments display a consistent increasing trend in correlation coefficients as the backward calculation time
increases. For instance, in LAG (SAGE III/ISS), the correlation coefficients for the ASM region increase from 0.53 (with a 60-
day backward period) to 0.69 (with a 180-day backward period), and from 0.43 to 0.75 for the NAM. The most rapid increase
occurs when extending the backward period from 60 to 120 days. These significant improvements in the reconstruction suggest
that UTLS water vapor concentrations in August are partially influenced by processes from boreal spring or even winter, with
a potentially stronger influence at higher altitudes, as air parcels at those levels typically require longer backward periods to
trace back to the CPT.

To further investigate the control of lower stratospheric water vapor mixing ratios by the large-scale temperature field, we
now correlate the SAGE III/ISS and MLS water vapor values observed above the tropopause with the respective CPTs either
derived from the Lagrangian reconstruction or from the local temperature profiles. Figure 4 presents scatter plots illustrating
observed water vapor mixing ratios versus CPTs for the three experiments: Loc, Lag (SAGE III/ISS), and Lag (MLS). Specifi-
cally, the grey dots in Fig. 4a-c represent saturation values at local CPTs, while those in Fig. 4d-i represent the saturation values
at Lagrangian CPTs.

Consistent with the results in Fig. 3, the correlation between observed water vapor concentrations and local CPTs from LOC
(Fig. 4a-c) is very weak. The saturation values calculated using local CPTs (grey points) show large moist biases compared to
observed values: 15.14 ppmv on average in the tropics, 6.16 ppmv in the ASM, and 13.48 ppmv in the NAM. In contrast, the
reconstructions for LAG show only ~1-2 ppmv dry biases for all regions, significantly reducing the overall biases. Moreover,
the correlations between water vapor concentrations and Lagrangian CPTs are much stronger, ranging from 0.60 for the ASM
based on the SAGE III/ISS dataset (Fig. 4e) to 0.78 for the NAM region (Fig. 4f). Notably, the scatter plots for monsoon
regions (Fig. 4e, f, h, i) exhibit no significant differences in overall structure compared to those for the tropics (Fig. 4d, g). This
suggests that the primary control mechanism for UTLS water vapor in monsoon regions is likely the same as that for the entire
tropics.





The regression lines for observations versus Lagrangian CPTs (blue lines) in Fig. 4d-i all have smaller slopes than those for the saturation mixing ratios (grey lines), likely due to the influence of points above 19.0 km. The slopes of the regression lines for the ASM and NAM based on the SAGE III/ISS dataset (Fig. 4e-f) are more aligned with the saturation slopes, due to less sampling at high altitudes. As altitude increases, tracing air particles back into the troposphere requires a longer back-trajectory, which introduces more uncertainties in determining CPT. Additionally, air above 19.0 km is more likely to be well-mixed within the stratosphere, which explains why using climatological water vapor concentrations shows better consistency, as depicted in Fig. 2. Thus, instead of being correlated with CPT, water vapor concentrations above 19.0 km appear more related to the climatological moisture conditions within the stratosphere.

Through the discussion above, we assess the performance of the Lagrangian method in water vapor reconstruction from various angles: horizontal and vertical distributions, correlation between reconstructions and observations, and the relationship between altitude, observations and Lagrangian CPT. The overall distributions in both horizontal and vertical directions indicate the satisfactory performance of the Lagrangian CPT reconstruction, effectively capturing the main variations in water vapor concentrations in the boreal summer within monsoon regions. By comparing the results from LAG and LOC, we demonstrate significant advantages of using Lagrangian CPT over local CPT. The use of Lagrangian CPT leads to significantly improved correlations between simulated and observed mixing ratios and substantially reduced biases in determining water vapor concentrations. However, the Lagrangian method also has limitations. Calculating the back-trajectories of each cluster, particularly over a 180-day period for greater accuracy, requires considerable time and storage. More importantly, the results of LAG reveal common dry biases throughout the entire tropics and the UTLS layer. Horizontally, the reconstructions exhibit dry biases of 1-2 ppmv, particularly in the NAM region. Vertically, there are biases of 1-2 ppmv above 16.5 km (around the tropopause) and of 3-5 ppmv below, in the troposphere. Above approximately 19.0 km, climatological water vapor concentrations show greater consistency with observations compared to the reconstructions calculated using Lagrangian CPTs. Without accounting for effects of convection, the Lagrangian method tends to produce dry biased results. The reasonable representation of stratospheric moistening above monsoon regions suggests that using back-trajectories driven by reanalysis wind fields and temperature can, to some extent, determine stratospheric water vapor, even during convective seasons and in convective areas.

## 3.2 Locations of the Lagrangian Cold Points

The Lagrangian reconstruction not only allows for the reconstruction of observed water vapor values but also provides insights into the atmospheric regions where dehydration has occurred. Since relevant dehydration seems to have taken place weeks to months before the observation time, it is valuable to determine the regions of strongest dehydration and whether these regions are strongly localized or more homogeneously distributed across the tropics. Utilizing all back-trajectories (from LAG), we trace the observations back to the specific locations of their Lagrangian cold points. Given the large number of such trajectories, we calculate the spatial distribution of these locations using probability density functions (PDFs). The scatter plots of the Lagrangian cold point locations are shown in Fig. 5a-d (with colors denoting the reconstructed water vapor values), and the corresponding PDFs are presented in Fig. 5e-h.



The results from the SAGE III/ISS and MLS datasets show similar patterns for both monsoon regions. In the ASM region (Fig. 5a-b), Lagrangian cold points are spread across the 0-30°N zonal band, with most dehydration points situated in the ASM region and some extending into North Africa and North America. According to the PDF of the Lagrangian cold points in Fig. 5e-f, most of the Lagrangian cold points are located over India and the Bay of Bengal, around 10°-30°N, 70°-95°E, indicating the primary origin of water vapor in the ASM. The top 10% of the highest reconstructed water vapor concentrations (exceeding ~6 ppmv) are concentrated in the same region (red contour lines), slightly displaced towards higher latitudes. This suggests that the increased water vapor in the ASM is primarily attributed to dehydration processes occurring in the vicinity of the monsoon over Asia. The backward time length required for air parcels to reach these Lagrangian cold points is shown in Fig. S2a-b, indicating that the dehydration processes occur over a timescale of days before the air parcels reach the observation points. Other Lagrangian cold points, located further away and with lower reconstructed water vapor concentrations (1-5 ppmv), correspond to longer time periods (1-6 months) between the dehydration event and observation. While these low water vapor air parcels are not the primary factor for the monsoon moist anomalies, their contribution to the final reconstructions highlights the need to extend the simulated backward time period, especially considering the improvements in correlation coefficients shown in Fig. 3.

For the NAM region (Fig. 5c-d), significant scatter is observed across North America, with overall patterns extending throughout the 0-30°N zonal band, including into South Asia. Surprisingly, the PDFs in Fig. 5g-h indicate that the primary dehydration center is in the ASM region, meaning that most air parcels in the NAM UTLS experienced dehydration in South Asia. Focusing on the top 10% highest reconstructed water vapor concentrations (Fig. 5g-h), we identify two leading centers for the Lagrangian cold points. One center remains in South Asia, a similar region to the ASM dehydration center but slightly southeastward. The other, more significant center is near the NAM itself, likely the main contributor to the increase in reconstructed water vapor concentrations in the NAM. This suggests that the increase in reconstructed water vapor concentrations in the NAM region is primarily influenced by local tropopause temperatures, with additional moisture contributions from transport from South Asia. In the trajectory simulations, the average backward period required to trace observed air parcels back to their Lagrangian cold points for the NAM is ~45 days (Fig. S2c-d). This indicates that the temperatures used to reconstruct the water vapor at those Lagrangian cold points are partially from June or even earlier, which are lower than the temperatures in August, leading to lower reconstructions.

In summary, the analysis of Lagrangian cold points along back-trajectories reveals significant insights into the distribution of the cold temperature regions and corresponding dry points that contribute to UTLS water vapor within the monsoon regions. Specifically, regions over India and the Bay of Bengal are crucial for the water vapor budget in the ASM. For the NAM region, both local sources in the Western Pacific and Gulf of Mexico, as well as sources over South Asia, determine the final water vapor values. The combination of long-range transport from remote regions and local convection—particularly, with remote influences being more substantial for the NAM than the ASM—appears to be decisive for the final moisture composition within the anticyclones. A remaining question is how the Lagrangian reconstruction resolves the contribution of local convection, which will be discussed in the next section.



# 4   Discussion: Lagrangian reconstruction and convection

To further investigate the cause of the common dry biases and the related effects of convection, we follow recent studies (e.g. Randel et al., 2015; Peña-Ortiz et al., 2024) and use OLR as a proxy for convection. We specifically analyze the potential influence of convection on the Lagrangian water vapor reconstruction. Randel et al. (2015) used MLS observations from May to September (2005-2013) to obtain time series of UTLS (100 hPa) water vapor concentrations above the ASM, separating specific wet and dry phases to reveal the corresponding anomalous OLR patterns. Their findings indicate that OLR anomalies exhibit a dipole structure over the ASM region. The decrease in OLR anomalies (indicating strong convection) over the eastern part of the dipole (20-30°N, 80-110°E) corresponds to the dry phase (i.e. low UTLS water vapor mixing ratios over the ASM) and vice versa. We conduct a similar analysis to derive OLR indices and then composite water vapor concentrations for observations from SAGE III/ISS and the reconstructions. Two OLR indices are defined according to the dipole structure—OLR-West and OLR-East (Sect. 2.2)—and are used to select days with high-OLR ($\geq$ 1.5 standard deviations) and low-OLR ($\leq$ -1.5 standard deviations). The west-east shifts in convection, as reflected in these OLR indices, may be related to different modes of the ASM anticyclone (Honomichl and Pan, 2020).

Figure 6 shows water vapor observations and reconstructions averaged over the 0–10 days following high and low OLR events. Note that anomalously low OLR corresponds to increased convection, and vice-versa. Consistent with Randel et al. (2015), using the OLR-East index shows that observed water vapor mixing ratios below 16.5 km (~100 hPa), composited for low-OLR days (strong convection), are dryer than those for high-OLR days, with the highest difference of 6.6 ppmv at 15.5 km (Fig. 6a). The profiles using the OLR-West index show the opposite results, with increased moisture for low OLR corresponding to strong convection (Fig. 6b). Our results confirm that strong convection in the eastern part of the ASM is associated with drying of the UTLS, while the strong convection shifting to the western part is associated with moistening of the UTLS.

The right panels of Fig. 6 show the results for reconstructed water vapor profiles. Using either the OLR-East or OLR-West index does not significantly affect the reconstructions or the differences between strong and weak convection periods. This suggests that neither the west-east shift nor the intensity of convection has a substantial impact on the reconstructions. This finding indicates that the simple Lagrangian water vapor reconstruction method fails to capture the moistening and drying processes associated with convection and ice injection in the monsoon regions. This limitation may be due to ERA5's inadequate representation of temperature variations associated with convection variability or with the missing representation of ice injection and microphysics in the simplified dehydration method. This limitation could also be a primary cause of the common dry biases in Lagrangian reconstructions and the limited coverage of reconstructed water vapor anomalies over the ASM, as compared with observations in Fig. 1e and g.

Additionally, the water vapor reconstructions for the NAM region are less satisfactory compared to the ASM. In the NAM region, the anomalies are only partially reproduced, while the ASM anomalies are nearly fully captured (Fig. 1c-d, g-h). Our results show that the correlation coefficients between the Lagrangian CPTs and observed water vapor concentrations are highest for the NAM region (Fig. 4f and i). However, despite this strong correlation, the reconstructions based on the Lagrangian CPTs





perform less satisfactorily. This discrepancy, along with the biases observed in the NAM region, could be linked to several factors: errors in representing local temperature variability and convection, or inaccuracies in transport processes. Homeyer et al. (2024) suggest that the processes driving stratospheric hydration during NAM convection often involve ice sublimation without significant changes in other trace gases, indicating a unique characteristic of NAM events that may not be captured adequately in standard reconstructions. Additionally, as suggested by our trajectory simulation results, the long-range transport from South Asia to the NAM region appears to significantly influence NAM water vapor concentrations, meaning that errors in ERA5 wind and transport processes could also contribute to these biases. Homeyer's findings emphasize the role of multiple, competing mechanisms within convection events, which may complicate the representation of long-range transport effects in models. Further investigation of other tracers originating from Asia could help to clarify whether the long-range transport from Asia to the NAM region and its remote influence on NAM water vapor levels is accurately represented.

## 5 Conclusions

This study investigates the performance of Lagrangian reconstructions of UTLS water vapor during the boreal summer monsoon seasons over Asia and North America. Our results demonstrate the effectiveness of the Lagrangian method in representing UTLS water vapor variations and structures during boreal summer. Compared to traditional methods using local CPTs, which overlook the horizontal movement of air parcels, the Lagrangian method significantly reduces biases in water vapor concentrations and improves correlations with observations across the tropics and monsoon regions.

Despite a systematic dry bias of 1-2 ppmv in the Lagrangian reconstruction, the simulation successfully captures the moist anomalies in the ASM during the boreal summer. This indicates that UTLS water vapor concentrations in the ASM are primarily influenced by large-scale tropical tropopause temperatures and a similar freeze-drying mechanism as in the deep tropics.

However, the Lagrangian method fails to reproduce the moistening in the NAM region, which may be related to temperature or transport biases in the ERA5 reanalysis data. Notably, ERA5-driven calculations show that UTLS water vapor mixing ratios in the NAM are significantly influenced by transport from South Asia and associated tropopause temperatures, while the highest mixing ratios are mainly controlled by local tropopause temperatures. The larger simulation biases in the NAM region could be due to a limited representation of local convection over North America in ERA5 or inaccuracies in the simulations of remote transport from Asia.

Additionally, the Lagrangian method does not capture the drying and moistening effects of east-west convection shifts in the ASM as proposed by Randel et al. (2015), likely due to unresolved temperature variability in ERA5 and the absence of ice microphysics in our simplified trajectory calculations.

Overall, our study highlights the Lagrangian method's capability to enhance the understanding of UTLS water vapor variability and provides valuable insights into climate feedback mechanisms. Future research should address remaining challenges, such as improving temperature accuracy in reanalysis data and incorporating missing moistening/drying processes (e.g., convection), to further refine simulation approaches for a more comprehensive representation of UTLS water vapor.



**Figure 1.** Horizontal distribution of water vapor concentrations and anomalies in August. Observed water vapor concentrations (a-b), the reconstructed concentrations (c-d) and corresponding anomalies (e-h) based on SAGE III/ISS at 16.5 km (left) and MLS at ~ 16.3 km (right). Grey boxes in each subplot show the defined area of ASM (15°-35°N, 50°-150°E) and NAM (10°-35°N, 160°-80°W). Reconstructions in this figure use both TSTs and non-TSTs, the portions of TST are shown with upper right strings of c-d and g-h.



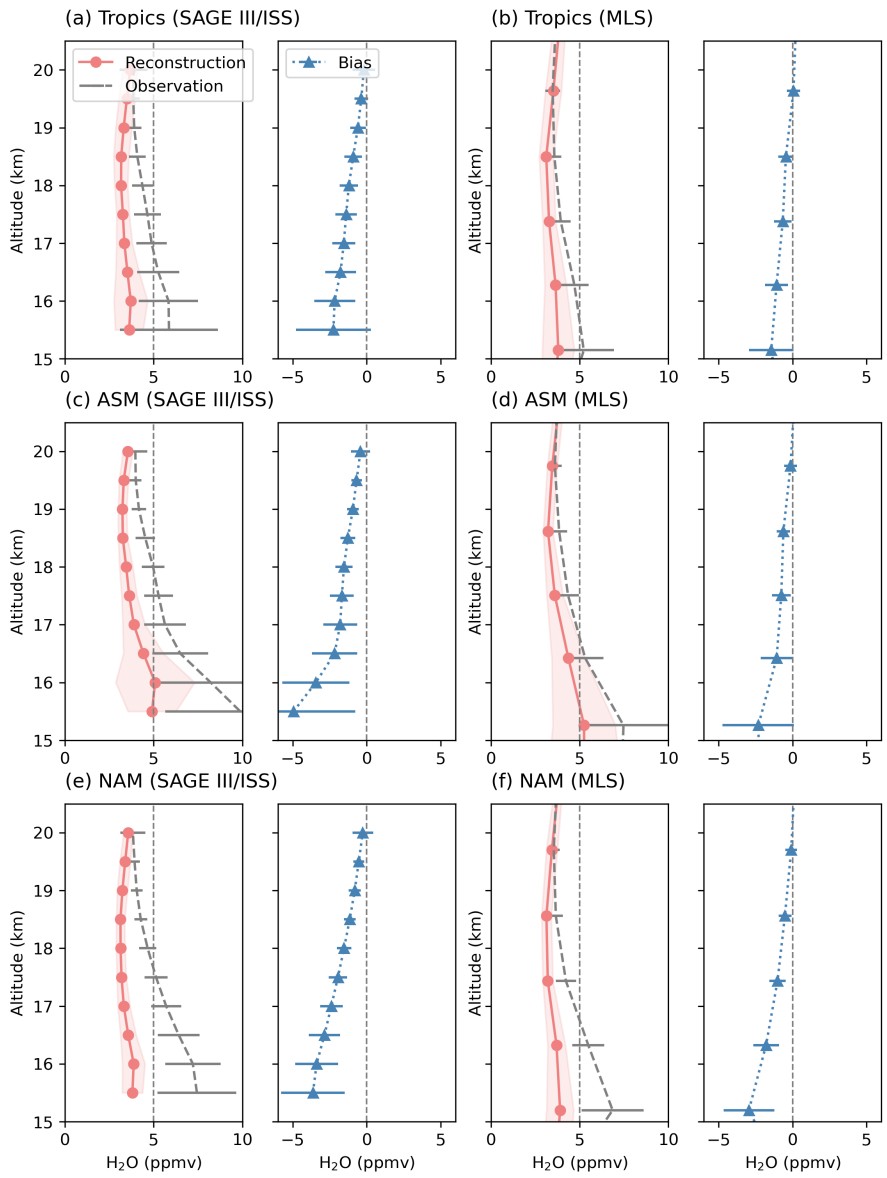

**Figure 2.** Vertical profiles of water vapor concentrations in August. For each subplot, it shows observed water vapor concentrations (grey dotted lines), reconstructed concentrations (red lines, including both TSTs and non-TSTs), and the bias between them (reconstructed values substract observed values, blue lines). Upper, middle and lower columns show the averaged values in tropics (35°S-35°N), ASM and NAM, from SAGE III/ISS (left panels) and MLS (right panels).

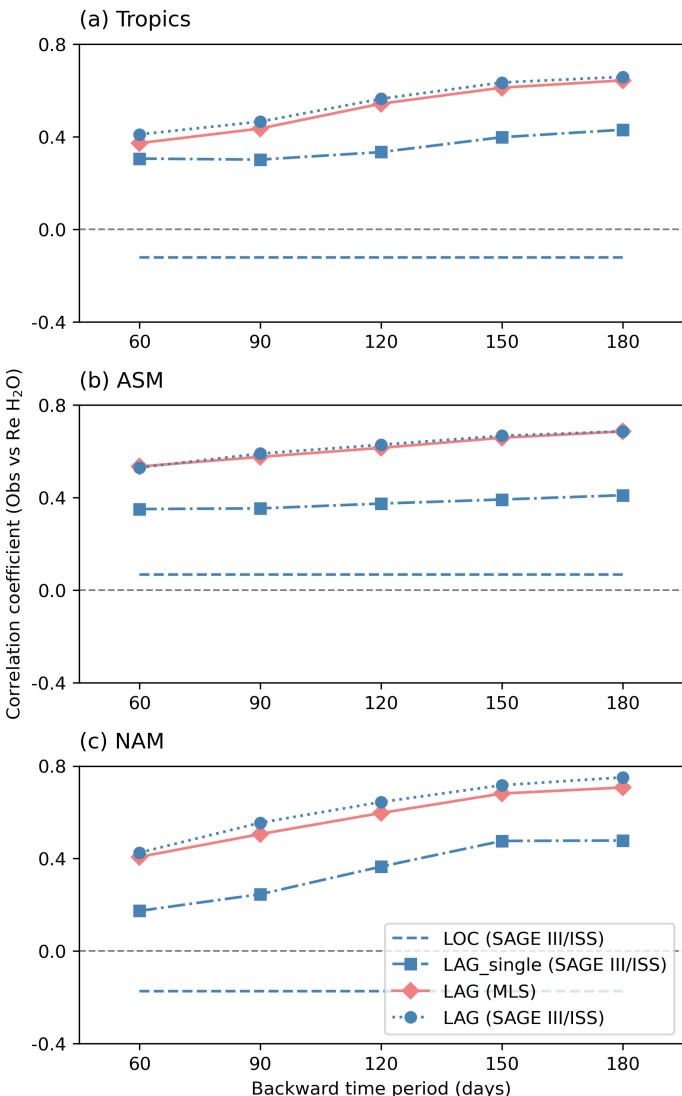

**Figure 3.** Correlation coefficients between observed and reconstructed water vapor concentrations (TST-only). Upper, middle and lower panels show the correlation coefficients between 15.5-20.0 km within in entire tropics (a), ASM (b) and NAM (c), respectively. Red diamonds represent the results of Experiment LAG based on MLS dataset. Blue crosses, squares and rounds represent the results based on SAGE III/ISS dataset of LOC, LAG_single and LAG, respectively.



**Figure 4.** Scatters of water vapor concentrations (TST-only) vs dry point temperatures. Left: water vapor concentrations from SAGE III/ISS vs local dry point temperatures (Experiment Loc). Middle: water vapor concentrations from SAGE III/ISS vs Lagrangian dry point temperatures (Experiment LAG based on SAGE III/ISS). Right: water vapor concentrations from MLS vs Lagrangian dry point temperatures (Experiment LAG based on MLS). Coloured points indicate the observed water vapor concentrations with the colour showing altitudes of the points. Grey dots represent reconstructed concentrations (saturation).





**Figure 5.** Horizontal distributions of the locations of the Lagrangian cold points (LCPs) used for water vapor reconstruction at 16.5 km derived from Experiment Lag and their probability density functions (PDFs). The locations of the LCPs are shown with colors representing the reconstructed water vapor concentrations, with starting points in ASM (a, b) and NAM (c, d). The scatters are plotted in ascending sequence according to the values of reconstructions. The blue contour lines in a-d represent the CPTs at 192 K. The PDFs of these locations are presented for ASM (e, f) and NAM (g, h), and the red contour lines in these plots represent the PDFs of the locations with the top 10% highest reconstructed water vapor concentrations. The left panels (a, c, e, g) show results based on SAGE III/ISS, while the right panels (b, d, f, h) show results based on MLS data. The black boxes indicate the original regions of starting points.

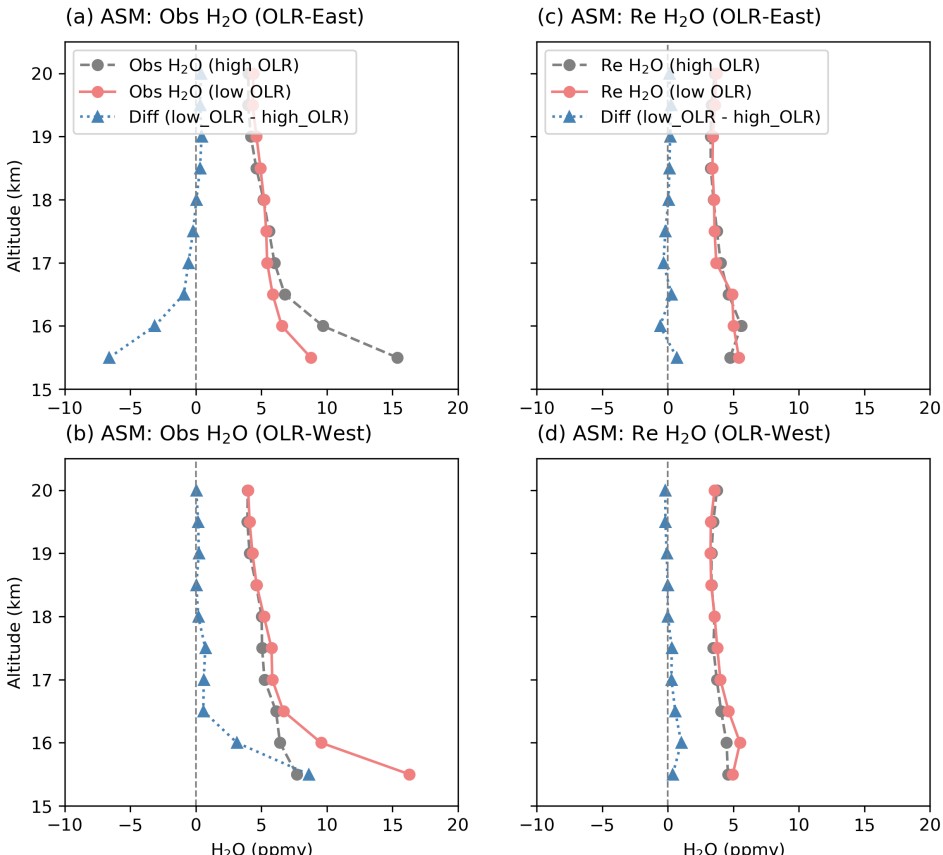

**Figure 6.** Vertical profiles of water vapor concentrations under the influence of convention within the ASM region, based on SAGE III/ISS dataset. The left panels show observed water vapor profiles averaged during high-OLR days (weak convention), low-OLR days (strong convection), and the difference, with OLR indices averaged within the eastern part (a) and western part (b). Right panels show reconstructed water vapor profiles averaged during high-OLR days, low-OLR days, and the difference, with OLR indices averaged within the eastern part (c) and western part (d).



*Code and data availability.* The CLaMS model is available in the CLaMS git database. Detailed information is available at https://clams. icg.kfa-juelich.de/CLaMS/GitLabInstructions. ERA5 reanalysis data are available from the European Centre for Medium-range Weather Forecasts (https://apps.ecmwf.int/data-catalogues/era5/?class=ea), last access: 03 August 2024). The MLS v5.0 water vapor data used in this study are available from NASA's Earthdata website (https://www.earthdata.nasa.gov/learn/find-data/near-real-time/mls). SAGE III/ISS
Level 2 Solar Event Species Profiles (HDF5) Version 5.3 data can be accessed through NASA's Atmospheric Science Data Center (https: //asdc.larc.nasa.gov/project/SAGE%20III-ISS/g3bssp_53). The NOAA CPC OLR data are available at (https://psl.noaa.gov/data/gridded/ data.cpc_blended_olr-2.5deg.html).

*Author contributions.* H. W. carried out the analysis and wrote the original draft of the manuscript. P. K. and F. P. supervised the research, contributing ideas, guidance, and discussions throughout the study, and assisted with iterative revisions. M. P., M. T., C. P., and N. P. provided
comments and suggestions during the manuscript revision. All authors contributed to discussions and final revisions of the paper.

*Competing interests.* The authors declare no competing interests.

*Acknowledgements.* The authors would like to express their gratitude to the European Centre for Medium-Range Weather Forecasts (ECMWF) for providing meteorological analysis for this study. We extend our appreciation to Nicole Thomas for her exceptional programming support. Additionally, we thank ChatGPT (https://chat.openai.com, last accessed: 2 October 2024) for their assistance in refining the final text. The
CPC Daily Blended Outgoing Longwave Radiation (OLR) - 2.5 degree data was kindly provided by the NOAA PSL, Boulder, Colorado, USA, via their website at https://psl.noaa.gov.



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
