# Peer review of "Understanding Boreal Summer UTLS Water Vapor Variations in Monsoon Regions: A Lagrangian Perspective"

_EGUsphere, 2024_

## Referee Comment (RC3)

"Understanding Boreal Summer UTLS Water Vapor Variations in Monsoon Regions: A Lagrangian Perspective"
egusphere-2024-3260

**Summary:**

This paper uses Lagrangian back-trajectories calculated from ERA5 meteorological fields to reconstruct the horizontal and vertical distributions of water vapor in the tropical upper troposphere and lower stratosphere (UTLS) during the boreal summer. The paper poses three questions. First, how do reconstructed UTLS water vapor distributions compare to co-located measurements from SAGE-III ISS and MLS? Second, are moisture anomalies controlled locally or remotely by the Lagrangian (upstream) CPT? Finally, do the Lagrangian reconstructions agree with the finding of Randel et al. (2015) that stronger convection (in the Asian Summer Monsoon or ASM and the North American Monsoon or NAM) leads to drier air moving into the stratosphere?

The reconstructions are carried out using the CLaMS trajectory module (Konopka et al., 2022). Back trajectories are calculated for 180 days using the CLaMS model's trajectory module and were initiated from the satellite measurement locations and times. Results are presented for comparisons of the reconstructions with SAGE-III and MLS water vapor values at the back-trajectory initiation points. The water vapor reconstructions are based upon the coldpoint temperatures, with the coldpoints identified either from the local vertical temperature profile or from the back-trajectory minimum temperature (the Lagrangian CPT). Three types of reconstructions are thus done based on the type of CPT: (a) using local CPTs (LOC), (b) using the Lagrangian CPT for every single trajectory (LAG_single), and (c) using the average Lagrangian CPT for a cluster of 51 back trajectories.

From their results the authors conclude that (a) the Lagrangian approach significantly improves upon approaches based upon local CPTs, (b) despite a dry bias, the Lagrangian reconstructions successfully capture the horizontal distribution of moist anomalies in the ASM but not in the NAM, but (c) the reconstructions do not capture the relative drying and moistening associated with east-west shifts of convection in the ASM that was observed by Randel et al. (2015).

**General and specific comments:**

This paper addresses a topic that has received considerable attention over the past 20 years or so. And a positive feature of the approach that the authors have undertaken in this work is the direct comparison of their water vapor reconstructions with SAGE-III ISS and MLS water vapor observations.

One aspect of the study that deserves more explanation are the significant low biases of the reconstructed water vapor mixing ratios relative to both the SAGE-III and MLS

observations that are evident in both Figs. 1 & 2.  Following Liu et al. (2010) they attribute the dry bias to "missing cloud microphysics", but that is not the end of the story.  Indeed, using domain-filling approach, the reconstructed water vapor fields 100 and 82 hPa obtained by Schoeberl, Dessler and Tao (2013) display very little bias with respect to MLS - without including any microphysics other than allowing for a limited degree of supersaturation at the LCPs.  In any case, I would recommend the authors include some commentary on this topic relative to the very interesting study of the dehydration occurring in StratoClim by Konopka et al. (2023) as this addresses the impact of microphysics on dehydration along CLaMS parcel trajectories.

One very interesting result is presented in Fig. 5.  It shows that while the vast majority of the Lagrangian cold points upstream of the observations in the ASM are within the ASM, the NAM is a very different story. Although a small fraction of the NAM LCPs come from the NAM region, the majority of the LCPs lie within the ASM.  This is an important finding since it emphasizes the dominant role of the Asian monsoon in controlling the moisture entering the stratosphere in boreal summer, while the North American monsoon is relatively speaking a bit player.  This result could well be highlighted more explicitly in the Conclusions section.

In Section 4, the authors address the finding of Randel et al. (2015). They are able to repeat the Randel et al. results with the satellite water vapor observations but not with the reconstructed water vapor fields.  They argue that the simple Lagrangian fails to properly capture the effects of convection and ice injection in monsoon regions.  This is not a convincing argument given that Konopka et al. (2023) did not find that convective processes played a significant role in determining the final stratospheric water vapor entry values in the circulation around "dehydration carousel" in the Asian summer monsoon anticyclone.

As a general comment, I found the narrative flow of the text choppy and confusing, particularly in the Introduction.  The Introduction certainly recognizes the long-standing consensus that the dominant control on the concentration of water vapor entering is through slow horizontal transport.  However, this is restated in various ways multiple times, suggesting a controversy that does not exist (see comment #4 below). There are certainly many ramifications of this general principle, and indeed this paper explores some of those. At minimum, I would recommend a revision of the Introduction to make it shorter and read more smoothly

**Specific comments:**

1. I found the discussion of the methodology of the water vapor reconstructions (Section 2.3.3) is incomplete.   They have adopted is to do three types of reconstructions ("experiments"), two obtaining water vapor values from the CPT along back trajectories ("LAG_single" and "LAG") and a third that is based upon local CPTs ("LOC").  The first two types of reconstructions appear to be similar in

approach to the Lagrangian trajectories used in similar studies going back at least two decades [see, for example, Fueglistaler and Haynes (2005)]. However, the method of by which the "LOC" reconstructions are carried out is unclear, especially since it lumps all three of the reconstruction approaches in one paragraph.

2. It would have been helpful if the captions for Figs. 1 & 2 specifically stated that the reconstructions were obtained through the LAG "experiment".

3. (line 31) Pan et al. (2018) did not introduce the Lagrangian Cold Point, although they do provide a number of references to the Lagrangian approach to determining the effective dehydration temperature for air parcels entering the stratosphere. Of these, the oldest reference is to Fueglistar et al. 2004, although the exact phrase only appears in later papers such as Kruger et al. 2008, for example.

4. (L.25-36) Taken together these sentences, comprising the latter half of the first paragraph in the Introduction, restate the importance of the concept of the transport history of air parcels entering the stratosphere several times over. As the references attest, Lagrangian approaches have been used been for over two decades, but the paragraph gives little sense of what new insights the Lagrangian perspective has provided since the earlier papers such as Fueglistaler et al, 2005.

**Recommendation:**

Acceptance subject to minor revisions.

---

## Author Comment (AC1)

**Response to Referee 1**

We thank Referee 1 for the thoughtful and detailed feedback. We appreciate all comments which clearly helped to improving the manuscript, and we addressed all points in the revised version. Reviewer comments are in black, answers in green. The main changes in the revised version are:

- A thorough rewriting of large parts of the text, including the Abstract, to discuss results and relations to atmospheric processes in a much clearer way.

- A clearer discussion of the dry bias in the reconstruction, supported by the inclusion of tropopause height information.

- A new analysis examining the relationship between reconstruction bias and convection. This, along with the original discussion related to convection, has been reorganized and is now presented in Section 3.3.

General feedback: This paper aims to improve our understanding of lower stratospheric water vapor anomalies that occur over the Asian and North American summer monsoons, a problem that has implications for surface climate and stratospheric chemistry. This paper uses a Lagrangian trajectory method to identify the role of cold point temperatures in the vicinity of the monsoon in setting the water vapor content of air reaching the lower stratosphere. I believe that this is a valuable contribution that can be suitable for publication in ACP following revisions.

Comment 1: This work uncovers a correlation between Lagrangian cold point temperatures and water vapor anomalies over the Asian summer monsoon. However, the mechanism presented here can only explain a fraction of the overall water vapor anomaly. While the dry bias of the Lagrangian trajectory method has been noted before, the dry biases in Fig. 2 make it difficult to claim that elevated Lagrangian cold point temperatures contribute significantly to the water vapor anomalies. For example, at 15.5 km the ASM reconstructed anomaly is about 1 ppm, while the SAGE anomaly is about 5 ppm. Therefore, the current method only accounts for about 1/5 of the observed moistening in the ASM. In the NAM, the Lagrangian method does not show a moistening. In both regions, I feel that the current presentation of these results overstates the moistening that can be explained by this method. This framing needs to be improved prior to publication.

We agree with the reviewer that our Lagrangian reconstruction results in a dry bias and an underestimation of the monsoon moist anomaly. This is particularly so at lower levels, e.g. 15.5km. The main reason is that the Lagrangian reconstruction, which finds the Lagrangian cold point along the back trajectories, only works in the stratosphere, i.e., after the "true" Lagrangian cold point has been passed by an air parcel on its way from the troposphere to the stratosphere. This limitation is particularly relevant for altitudes below the cold point tropopause, such as the region over the ASM shown in Fig. 1 and the lower portions of the profiles in Fig. 2, where the reconstruction method cannot be expected to perform well, as these levels remain within the troposphere. Thus, we acknowledge that this constraint was not sufficiently emphasized in our earlier discussion. To better address this issue, we added the position of the climatological August cold point tropopause, as derived from ERA5, in Fig. 2. We also incorporated the observed moist anomalies in monsoon regions relative to the entire tropics (grey bars in the left sub-panels in Fig.2), along with the corresponding reconstructed anomalies (coral bars), to better evaluate the reconstruction's performance in capturing these features. Specifically, the reconstruction reproduces one-third of the observed anomalies in the ASM at 15.5 km, increasing to over two-thirds at 16.5 km and to an even higher contribution at levels above.

Comment 2: Moreover, I would argue that the central conclusion of this paper is that a small portion of moistening over the ASM is caused by an altered transport pathway through the UTLS, not that the moistening can be explained by the Lagrangian method. A secondary conclusion would be that the altered pathway is not significant for the NAM. In other words, the ASM allows some portion of air to avoid the "cold trap" and the dehydration that would occur within. This results in a water vapor anomaly that occurs regardless of direct injection of water vapor/ice into the lower stratosphere (although the majority of the anomaly is driven by these other processes). The correlation between the Lagrangian reconstructions and ASM observations suggest that this cold-trap-avoidance mechanism is robust, but it does not prove that the mechanism is the dominant source of water vapor anomalies.

We agree that the Lagrangian reconstruction method does not explain the entire anomalies observed by SAGE and MLS. However, the method works reliably well in the ASM region to reproduce the pattern of anomalies (Fig. 1), and also reproduces the main part of the anomaly at levels from the tropopause upwards (please refer to our reply to comment 1, and Fig. 2). However, since the number of TST trajectories decreases and the impact of the climatology on

the reconstruction increases at higher levels, we agree that these arguments should be considered carefully. Now in Fig. 2, instead of a mix of TST-based reconstructions and climatological values, we use the reconstructions only from TSTs, so that we can see a more 'systematic' dry bias. Nevertheless, the new figures of the reconstruction's contribution to monsoon anomalies in Fig. 2 and the high correlation coefficients shown in Fig. 3 show that in the ASM a significant part of the anomalies can be explained by the Lagrangian reconstruction method. For the NAM, we agree that the simple method does not capture the moistening processes sufficiently. We have included more detailed discussion about the capability of the Lagrangian method to reconstruct moist anomalies in ASM and NAM regions, and in particular we critically discuss the failure of the method in reconstructing the NAM anomaly now in Section 4.

Comment 3: The proposed mechanism would also gain meaning with additional discussion of other water vapor sources. For example, Smith et al. (2017) studied a summertime water vapor enhancement over North America and found that frequent deep convection can deliver water vapor to the lower stratosphere. O'Neill et al. (2021) also provide a mechanism by which water vapor injection occurs over intense convection. Studies like these would explain why the hydration captured by the Lagrangian trajectory method is smaller than the observed hydration, especially over the NAM.

Yes, we agree that deep overshooting convection plays an important factor in the anomalies in the monsoons, in particular over North America. We now incorporate a discussion (in section 4) of the results from Smith et al. (2017) and O'Neill et al. (2021), and also Homeyer et al. (2024) who demonstrated that frequent deep convection over North America and intense convection can deliver water vapor to the lower stratosphere. In particular, the new analysis on the relation between the reconstruction bias and the intensity of convection in Sect. 3.3 (suggested by the other reviewer) shows clearly that convective moistening is one of the key processes missed by the simplified Lagrangian method.

Comment 4: Additionally, the choice of the 6-hr resolution needs to be justified for two reasons. First, the monsoon can act on timescales shorter than 6 hours, so it is possible that the Lagrangian trajectories do not fully capture the effect of the monsoons. Li et al. (2020) found that the improved temporal resolution of ERA5 led to more rapid transport than ERA-i, so it is possible that the 6-hr data used here does not fully capture convective transport. Second, it has been shown that trajectories calculated with 6-hr data have transport errors and warm

CPT biases relative to those calculated with 1-hr data (Pisso et al., 2010; Bourguet and Linz, 2022). It is possible that the warm CPT biases cancel out when calculating anomalies, but it is also possible that the anomalies calculated with 6-hr data are larger than those that would be calculated with 1-hr data. This would mean that the mechanism presented here is actually smaller than these results would suggest.

We agree that enhancing the temporal resolution of the data could make the freeze-drying effect stronger in our calculations. However, generating the necessary high-resolution data in space and time and recalculating all trajectories is a major computational effort and beyond the scope of this paper. In the revised version, we discuss these issues in more detail in the last paragraph of the Section 4 as suggested by the reviewer, and with reference to the suggested publications. We also acknowledge that higher temporal resolution, rather than spatial resolution, has greater potential to refine the representation of Lagrangian dry points and deep convection.

Comment 5: I would also advise moving the LAG_single comparison to the Supplemental. It is well known that single trajectories are not meaningful and that ensembles should be instead. As currently presented, the comparisons with LAG_single distract from the main results. I also feel that the MLS results could also be moved to the Supplemental to improve the focus on the comparison between reconstructed and observed water vapor. (The same conclusions are drawn when comparing reconstructions with MLS and SAGE.)

Thank you for your thoughtful suggestions. We acknowledge that single trajectories are generally less meaningful than ensemble-based approaches. However, we would like to clarify that even in the LAG_single experiment, the results presented (e.g., in Fig. 3) are not results of single trajectories but are averages over large trajectory ensembles. The key difference lies in how these ensembles are generated: in the LAG_single experiment, only one trajectory is initialized at each measurement point along the profile, whereas in the experiment LAG, a full ensemble of trajectories is initialized at each individual measurement point. Therefore, all presented results are derived from substantial trajectory ensembles, with the ensemble size varying between experiments. We appreciate your comment highlighting this potential source of misunderstanding and have clarified these points in the revised manuscript. Regarding the MLS results, we believe that keeping them in the main text is essential because SAGE III/ISS has much lower sampling compared to MLS. By presenting the MLS-based comparisons alongside SAGE III/ISS, we provide a stronger foundation for our conclusions and enhance the robustness of our analysis.

**Minor Comments**

Comment 1: Lines 2–4: "The amount of water vapor entering the stratosphere is sensitive to cold point temperatures, making NH summer monsoons more favorable for transporting water vapor into the stratosphere." Water vapor enhancements over Northern Hemisphere summer monsoons do not follow from elevated cold point temperatures. Deep convection can lower the CPT, so this statement needs to be clarified.

Thank you for pointing this out. We have revised the sentence to clarify the relationship between cold point temperatures and water vapor transport in monsoon regions. Additionally, the abstract now is thoroughly rewritten, with consideration of all the comments related to the abstract.

Comment 2: Line 5: "investigating their contributions to stratospheric water vapor." To my understanding, the water vapor reconstructions in this work are confined to the tropical lower stratosphere, and there is no evaluation of how monsoon water vapor anomalies contribute to the stratospheric water vapor budget. Therefore, I would avoid saying that the contributions to stratospheric water vapor are evaluated here.

Thank you for the suggestion. We agree and have revised the sentence to more accurately reflect our study's scope.

Comment 3: Lines 9–11: "it effectively reconstructs UTLS water vapor (correlation coefficient 0.75), capturing moist anomalies in the ASM, but performing less well in the NAM." Following from general feedback above, the high correlation here does not mean that the Lagrangian method can explain the magnitude of the water vapor anomalies.

Please refer our response to major comment 1 and 2.

Comment 4: Line 17: "The water vapor" -¿ remove "The."

Thank you for pointing out. It has been removed.

Comment 5: Line 25: "Large-scale vertical transport enhances lower stratospheric water vapor..." Is this meant to say that large-scale vertical transport spreads convectively-injected water vapor? Large-scale transport on its own does not enhance lower stratospheric water vapor.

Please refer to the new abstract.

Comment 6: Line 33: "Several studies have successfully reconstructed UTLS water vapor using Lagrangian methods..." I would argue that studies are able to capture UTLS water vapor anomalies, not total water vapor concentrations (e.g., Smith et al., 2021; Bourguet and Linz, 2022). This is an important distinction given the uncertainty surrounding the dry bias in Lagrangian reconstructions.

We agree with this distinction and have revised the sentence to make it more precise.

Comment 7: Lines 78–79: "within both Asian monsoon and North American monsoon regions" - tropical water vapor is also considered. Could be easier to say "across the tropics."

It has been revised according to your suggestion.

Comment 8: Lines 96–98: Clarify that 1-2-1 vertical smoothing is done on 1 km grid. (It's not currently clear if smoothing is done on 0.5 km grid or 1.0 km grid.)

We appreciate this point and have clarified in the text that the 1-2-1 vertical smoothing is applied on a 0.5 km grid.

Comment 9: Line 125: SAGE vertical resolution is reported as 0.5 km here. When is 0.5 km used, and when is 2.0 km vertical resolution used?

Thank you for pointing this out. We have clarified in the text that the native SAGE III/ISS vertical resolution is 2.0 km, but the profiles are retrieved on a 0.5 km vertical grid. This distinction is now explicitly stated.

Comment 10: Section 2.3: How many trajectories are calculated in total, and how many are with the ASM and NAM, respectively?

The relevant information is provided in Section 2.3: For each August from 2017 to 2022, SAGE III/ISS recorded 149, 203, and 2292 profiles for the ASM, NAM, and tropics, respectively. Taking the ASM as an example, the number of calculated trajectories is given by $149 \times 10$ (profiles $\times$ levels) for LAG_single and $149 \times 10 \times 51$ (profiles $\times$ levels $\times$ ensemble trajectories) for LAG. Accordingly, the total number of calculated trajectories for LAG_single is 1490 for the ASM, 2030 for the NAM, and 22,920 for the tropics, while for LAG, these numbers increase to 75,990, 103,530, and 1,168,920, respectively. For each August from 2017 to 2019, MLS provides 7801, 10,223, and 126,981 profiles for the ASM, NAM, and tropics, respectively. Since MLS calculations use five vertical levels, the number of trajectories for LAG_single is computed as $7801 \times 5 = 39,005$ for the ASM, $10,223 \times 5 = 51,115$ for the NAM, and

$126,981 \times 5 = 634,905$ for the tropics. For LAG, these values are further multiplied by 51, yielding the final trajectory counts.

Comment 11: Lines 171ff: "The large-scale patterns in the reconstructions are consistent with the observations...." I would argue that this is misleading, even with the acknowledgement of the dry bias that follows. The NAM anomaly is not present in reconstructions, so broad statements about large-scale patterns should be avoided. Similarly, in the following paragraph, the assertion that the "reconstruction captures the enhancements in water vapor concentrations and their locations" is misleading. The quality of the water vapor reconstructions in the ASM and the NAM should be discussed separately to avoid conflating the two.

Thank you for your comment. We agree that the statement was too broad and have revised it to more accurately describe the differences in reconstruction performance between the ASM and NAM. We also clarify that the quality of the reconstructions differs between the two regions and discuss them separately in the revised text.

Comment 12: Line 188: Specify that the cyan squares are in Fig. S1.

This point has been modified.

Comment 13: Lines 249–252: Please be more specific with what you mean by "exhibit no significant differences in overall structure" and explain how this suggests that the tropics and monsoon regions have the same primary controlling mechanisms. To me, the tropical scatter plots (Fig. 4d, g) appear qualitatively different than the NAM scatter plots (Fig. 4f, i). Also, although the ASM scatter plots appear more similar to the tropical scatter plots, these plots only consider the relationship between the CPT and observed water vapor, so it is possible that other mechanisms could contribute to the two separately.

We appreciate the reviewer's insights. In our revised text, we clarify that the similarity between monsoon regions and the deep tropics refers specifically to the correlation between lower stratospheric water vapor mixing ratios and LCP temperatures, not local cold point temperatures. This suggests that in both cases, water vapor is controlled by remote dehydration processes rather than local conditions, and that advection through the large-scale temperature field can explain water vapor mixing ratios above the tropopause in the ASM similarly well as in the deep tropics. While the scatter plots in Fig. 4 show some qualitative differences, the key takeaway remains that lower stratospheric water vapor in the ASM region, like in the deep tropics, is primarily governed by historical dehydration at LCPs. In the NAM region, we fully agree that

the simplified Lagrangian reconstruction method does not explain the observed mist anomaly - and this is discussed in a more balanced way in the revised manuscript in Sect. 4 (see also our reply to comment 2). We acknowledge that additional mechanisms may contribute to differences, but our findings emphasize the dominant role of large-scale transport and freeze-drying in shaping water vapor distributions.

Comment 14: Lines 261–275: I would advise either removing this paragraph or moving it to the Discussion or Conclusion section.

Thank you for the suggestion. We have moved this paragraph to the Discussion section for better alignment with the overall flow of the paper.

Comment 15: Section 3.2: I suggest including a panel to Fig. 5 with the location of the LCP for all tropical trajectories. This would allow for a comparison of CPT locations between the monsoon regions and the tropics, which would support the idea that the monsoons alter the transport pathways through the UTLS.

We agree that this addition would be helpful. We have added a panel to Fig. 5 showing the LCP locations for all tropical trajectories and updated the discussion accordingly.

Comment 16: Lines 291–293: "This suggests that the increased water vapor in the ASM is primarily attributed to dehydration processes occurring in the vicinity of the monsoon over Asia." This needs to be clarified. The location of the highest reconstructed water vapor concentrations suggests that the increased water vapor captured by the Lagrangian reconstruction (about 1/5 of the observed increase) is primarily driven by changes to transport near the ASM. The remaining 4/5 of the observed anomaly is attributable to other processes.

Please refer to our response to Major Comment 1.

Comment 17: Lines 316–318: The contribution of distant CPTs to the reconstructed NAM water vapor anomaly does not imply significance of distant CPTs to the observed anomalies. Instead, the relative inability of the NAM reconstruction to capture observed water vapor anomalies implies that local processes (e.g., direct injection of water vapor) are crucial for explaining the final moisture composition within the anticyclone. However, the Lagrangian method cannot reproduce observations, so you cannot draw objective conclusions about the behavior of the atmosphere with this method.

Thank you for pointing to that potential misunderstanding. We fully agree that in the NAM

other, local processes are likely important. Obviously, the related text was not clear. We have revised the text to clarify the role of distant CPTs and local convection in shaping water vapor anomalies in the NAM. The revised discussion now explicitly acknowledges that processes such as convective mixing and overshooting likely play a dominant role in enhancing water vapor in the NAM region, as the freeze-drying mechanism alone is insufficient to explain the observed anomalies (please also see our reply to major comment 2).

Comment 18: Line 351: "the ASM anomalies are nearly fully captured." The ASM observation and reconstruction patterns in Fig. 1 are qualitatively similar, but Fig. 2 shows that the ASM water vapor reconstruction does not capture a majority of the observed anomalies. Thus, I would caution against the quoted statement.

We have modified the statement to avoid this kind of unspecific comments.

Comment 19: Lines 370–372: Similar to the previous point, Fig. 2 shows that the Lagrangian water vapor reconstruction does not successfully capture the magnitude of the moist anomalies over the ASM. A portion of the moist anomalies can be explained by the Lagrangian reconstruction, but the large dry bias in the anomalies implies that a mechanism other than freeze-drying is needed to explain observations.

Please refer to our response to Major Comment 1.

**References**

Homeyer, C. R., Gordon, A. E., Smith, J. B., Ueyama, R., Wilmouth, D. M., Sayres, D. S., Hare, J., Pandey, A., Hanisco, T. F., Dean-Day, J. M., et al. (2024). Stratospheric hydration processes in tropopause-overshooting convection revealed by tracer-tracer correlations from the dcotss field campaign. *Journal of Geophysical Research: Atmospheres*, 129(16):e2024JD041340.

---

## Author Comment (AC2)

**Response to Referee 2**

We thank Referee 2 for the thoughtful and detailed feedback. We appreciate all comments which clearly helped to improving the manuscript, and we addressed all points in the revised version. Reviewer comments are in black, answers in green. The main changes in the revised version are:

- A thorough rewriting of large parts of the text, including the Abstract, to discuss results and relations to atmospheric processes in a much clearer way.

- A clearer discussion of the dry bias in the reconstruction, supported by the inclusion of tropopause height information.

- A new analysis on the relation between the bias in the reconstruction and convection, related to specific comment 5. This, along with the original discussion regarding the convection, has been reorganized and is now presented in Section 3.3.

General comment: This study employs a Lagrangian method to reconstruct water vapor over the Asian Summer Monsoon (ASM) and North American Monsoon (NAM) regions, investigating their contributions to stratospheric water vapor. While the introduction emphasizes that "In this study, we aim to further investigate the physical processes responsible for the enhanced water vapor over the ASM and NAM regions," I believe that the majority of the article primarily focuses on describing the distribution characteristics of water vapor and offers speculative suggestions regarding the underlying physical processes, rather than providing a thorough analysis of these processes and mechanisms.

Thanks for this critical general comment which shows us that overall our process analysis was not presented clear enough. However, we think that our paper indeed presents new insights into processes, and also into sources of biases for modelling approaches. In the revised version we tried to be much clearer in the description of processes and new results. In addition, we thoroughly addressed all specific comments below, which in our opinion helped to significantly improve the paper (in particular comment 5 - Thanks for that!). Therefore, we think that we could improve the paper so much that we can submit a revised and substantially improved new version.

Comment 2: The time periods for the MLS (August 2017 to 2019) and SAGE (August 2017 to 2022) datasets differ. Which time period was actually used in the study? In Figure 1, the differences in water vapor concentration between the two datasets—are these due to the different time periods being used? What is the rationale for using different time periods for the two datasets?

Thanks for the comment. We agree that the reason for the different periods needs to be explained. The extended period for SAGE was chosen to ensure sufficient coverage of the considered region. On the other hand, the large volume of MLS data makes it challenging to perform all trajectory calculations for a too long period - in particular because we launch large trajectory ensembles for each measurement point (we have included the specific numbers of trajectories that are calculated now in Section 2.3.1). However, if we confine the SAGE data to the MLS period, our results do not change significantly (see Fig. R1). We have included a short related discussion in the revised text (Section 2.1.2).

Comment 3: Fig. 1: The reconstructed water vapor over the NAM region does not reflect the observed features, such as the high water vapor values seen in the observations, unlike the ASM region. What is the underlying reason for this discrepancy? In the figure, the reconstructed water vapor uses the Lagrangian CPT method. If the tracing period were extended, for example, to 180 days as shown in Figure 3, would the reconstructed water vapor better capture the observed characteristics?

As Figures 1 and 2 use the maximum trajectory length of 180 days to determine the Lagrangian cold point, our finding that the reconstruction of water vapor above NAM is less accurate than over ASM is a robust result of our study. We think that this discrepancy is most likely due to the greater relevance of convective moistening events in the NAM compared to the ASM region which are not represented in the simplified Lagrangian reconstruction method. Also, the large circulation at around tropopause above the NAM shows stronger the year-to-year variability compared with that above the ASM (Park et al., 2007). A more thorough discussion of these issues is provided in the revised manuscript (Sect. 4).

Comment 4: Fig. 2: The reconstructed water vapor primarily relies on the CPT in the UTLS region. Why, then, do the differences in water vapor become smaller at higher altitudes?

At higher altitudes, the fraction of TST becomes smaller as backward trajectories need more than 180 days to reach the CPT, which is the maximum time of our backward calculations.

Hence, the fraction of non-TST trajectories, which are originating in the lower stratosphere and therefore represent climatological stratospheric mixing ratios. In this way, climatological mixing ratios start to dominate the reconstructed water vapor. A new related discussion paragraph is now included in Sect. 3.1. On the other hand, we agree that including climatological values in Fig. 2 could cause confusion, so we have revised the figure to show only the results based on TST trajectories. In the updated Fig. 2, we observe that in the deep stratosphere, the dry bias remains relatively constant at around 1.5 ppmv for both the entire tropics and the ASM. In contrast, the NAM again shows a different behavior, with a less consistent bias pattern, further indicating that the reconstruction performs less reliably for the NAM region.

Comment 5: Line 205: The authors attribute the discrepancies between the reconstructed and observed water vapor to issues with ERA5 temperature data or the absence of convective transport processes in the reconstruction model. This issue requires further analysis and diagnosis. If the difference between the observed and reconstructed results is calculated, does the time series of the difference align with the changes in the intensity of the convection (OLR)? If referring to the results in panels (b) and (d) of Figure 6, it seems likely that convective transport plays a significant role in the observed differences.

Thanks you for this insightful comment. Based on this suggestion, we have conducted further analysis and examined potential relations between the biases in reconstructed water vapor concentrations at 16.5 km and the time series of OLR indices as a proxy for convective intensity, as shown in Fig. R2. Our results indicate a significant correlation between the bias in the reconstruction and the OLR-West index, as a proxy for the strength of convection in the western monsoon region, particularly after applying a half-monthly mean to smooth out daily fluctuations. This new finding strongly supports that convection is a major influencing factor for the bias in the reconstruction. The analysis also reveals that convection in the western region has a stronger impact on reconstruction biases compared to the eastern region. This difference between the effect of convection in the western and eastern monsoon regions could have been expected given the result by Randel et al. (2015) that only convection in the Western region causes moistening of the UTLS. Hence, it is the moistening effect of convection which is likely underrepresented in the simplified Lagrangian reconstruction method and causing a large part of the reconstruction dry bias. Additionally, the correlation varies with altitude, with the strongest correlation observed at 16.5 km, indicating the complexity of atmospheric processes at different levels. To present this information in the revised manuscript we show the scatter

plots of the relation between the reconstruction dry bias and convective intensity in the new Fig. 8 and added a detailed discussion in Section 3.3 to specifically address this issue. We think that these new findings provide a much clearer relation between the bias in the reconstruction and the relevant atmospheric process of convection and significantly improve the paper.

Comment 6: Line 235: How should we interpret the influence of the minimum saturation mixing ratio (or the cold point temperature) from three months prior on the lower stratospheric water vapor in August?

This time lag is related to the water vapor tape recorder signal propagating upwards. Due to the very weak tropical upwelling, especially during the boreal summer (on the order of less than 0.5 K per day in the TTL), the water vapor values imprinted at the Lagrangian cold point propagate upward. Additionally, as mixing is weak, this memory effect propagates along the upward-moving trajectories within the ASM anticyclone, also referred to as the upward spiraling (Vogel et al., 2019). Of course, for air masses with Lagrangian cold points occurring before the onset of the monsoon (around beginning of June) the dehydration process is not related to the monsoon circulation. We have added some related text in the revised version to clarify this interpretation, in Section 3.1.2 where we discuss about the increasing of correlations.

Comment 7: Line 292: If the dehydration process occurs in the vicinity of the monsoon region, how can we understand that the internal region of the anticyclone over Asia acts as the upward pathway for the material?

According to Konopka et al. (2023), the ASM anticyclone acts as a major transport pathway for air ascending into the stratosphere, while dehydration predominantly occurs at its southern vicinity where air encounters the coldest temperatures near the tropopause. This process does not contradict the role of the anticyclone as an upward pathway; rather, it reflects the interplay between circulation and dehydration. As described in the "dehydration carousel" mechanism, air can recirculate within the anticyclone, lose moisture at its edges, and subsequently ascend into the stratosphere. We have included the related discussion in the second paragraph in Section 3.2.

Comment 8: Line 303: If the water vapor in the NAM region is transported from South Asia after undergoing dehydration, how is the moisture transmitted to the NAM region, and how does it form a high-value center in the NAM region (as shown in Figures 1a and 1b), given that the tropical summer region is dominated by an easterly wind belt?

The ERA5-based trajectory calculation clearly shows frequent transport from Asia to the NAM region via the subtropics, related to the westerly subtropical jet flow. However, these water vapor mixing ratios originating from freeze-drying (related Lagrangian cold points) over Asia are oviously too low and the NAM region in the reconstruction is significantly dry-biased. Hence, other processes not included in the reconstruction are likely responsible for moistening the NAM UTLS. The most likely process is local deep convection, which is known to frequently occur in the NAM region, though this process not necessarily involves direct injection into the stratosphere. An additional related discussion, including relevant recent literature has been added at the end of the Section 3.2 to address this point.

Comment 9: There are not many figures in the main text, so I suggest placing Figures S1 and S2 directly in the main text.

We appreciate this suggestion. However, we have added Fig. 8 in the main text, which is related to the comments 5, and have split the original Fig. 5 into Fig. 5 and 6 to include the locations of LCP for the entire tropics, allowing for a direct comparison with those in the monsoon regions. To maintain a balanced presentation and avoid overcrowding the main text, we will keep Figures S1 and S2 in the Supplementary Information.

[Figure]

Figure R1: Same as Fig. 1, but using SAGE III/ISS data from 2017–2019. The gaps in the left panels are due to limited sampling, where fewer than two profiles are available.

[Figure]

Figure R2: Time series of OLR indices and biases of reconstructed water vapor concentrations at 16.5 km for the ASM. The three panels correspond to results using OLR-East (a), OLR-West (b), and OLR averaged over both regions (c) as OLR indices. Blue bars represent daily OLR indices, while blue diamonds indicate half-monthly means. Red lines and squares show daily and half-monthly reconstruction biases based on SAGE III/ISS. The OLR indices are averaged over the 0–10 days preceding each date. Correlation coefficients between OLR and biases are shown in the legends, with a star indicating statistical significance at the 95% confidence level based on the Student's t-test.

**References**

Konopka, P., Rolf, C., Von Hobe, M., Khaykin, S. M., Clouser, B., Moyer, E., Ravegnani, F., D'Amato, F., Viciani, S., Spelten, N., Afchine, A., Krämer, M., Stroh, F., and Ploeger, F. (2023). The dehydration carousel of stratospheric water vapor in the asian summer monsoon anticyclone. *Atmospheric Chemistry and Physics*, 23(20):12935 – 12947.

Park, M., Randel, W. J., Gettelman, A., Massie, S. T., and Jiang, J. H. (2007). Transport above the asian summer monsoon anticyclone inferred from aura microwave limb sounder tracers. *Journal of Geophysical Research Atmospheres*, 112(16). Cited by: 280; All Open Access, Bronze Open Access, Green Open Access.

Randel, W. J., Zhang, K., and Fu, R. (2015). What controls stratospheric water vapor in the nh summer monsoon regions? *JOURNAL OF GEOPHYSICAL RESEARCH-ATMOSPHERES*, 120(15):7988–8001.

Vogel, B., Müller, R., Günther, G., Spang, R., Hanumanthu, S., Li, D., Riese, M., and Stiller, G. (2019). Lagrangian simulations of the transport of young air masses to the top of the asian monsoon anticyclone and into the tropical pipe. *Atmospheric Chemistry and Physics*, 19(9):6007–6034. cited By 64.

---

## Author Comment (AC4)

**Response to Referee 3**

We thank Referee 3 for the thoughtful and detailed feedback. We appreciate all comments which clearly helped to improving the manuscript, and we addressed all points in the revised version. Reviewer comments are in black, answers in green. The main changes in the revised version are:

- A thorough rewriting of large parts of the text, including the Abstract, to discuss results and relations to atmospheric processes in a much clearer way.

- A clearer discussion of the dry bias in the reconstruction, supported by the inclusion of tropopause height information.

- A new analysis on the relation between the bias in the reconstruction and convection. This, along with the original discussion regarding the convection, has been reorganized and is now presented in Section 3.3.

**General comments**

Comment 1: This paper addresses a topic that has received considerable attention over the past 20 years or so. And a positive feature of the approach that the authors have undertaken in this work is the direct comparison of their water vapor reconstructions with SAGE-III ISS and MLS water vapor observations. One aspect of the study that deserves more explanation are the significant low biases of the reconstructed water vapor mixing ratios relative to both the SAGE-III and MLS observations that are evident in both Figs. 1 & 2. Following Liu et al. (2010), they attribute the dry bias to "missing cloud microphysics," but that is not the end of the story. Indeed, using the domain-filling approach, the reconstructed water vapor fields at 100 and 82 hPa obtained by Schoeberl, Dessler, and Tao (2013) display very little bias with respect to MLS—without including any microphysics other than allowing for a limited degree of supersaturation at the LCPs. In any case, I would recommend the authors include some commentary on this topic relative to the very interesting study of the dehydration occurring in StratoClim by Konopka et al. (2023), as this addresses the impact of microphysics on dehydration along CLaMS parcel trajectories.

Yes, we agree that microphysical processes as proposed in Liu et al. (2010) are not the only ones that may explain the dry bias of the LDP water vapor reconstructions. As correctly pointed out by the reviewer, Schoeberl et al. (2013) demonstrated that including only a limited degree of supersaturation at LCPs significantly reduces the bias, without requiring more complex microphysical processes. This approach has also been applied in previous studies (Schiller et al., 2009; Ploeger et al., 2011), and we acknowledge that it provides a simplified yet effective representation of cloud processes. However, this simplification does not fully capture the complex cloud microphysics and mixing processes that influence stratospheric water vapor (Poshyvailo et al., 2018). Moreover, according to our results for the NAM, the reconstruction works not as it does for the ASM, showing dry bias beyond this systematic error. Additionally, the difference in dry bias between our study and Schoeberl et al. (2013) may also be influenced by trajectory sampling methods. Their study employed domain-filling forward trajectories based on ERA-Interim and MERRA meteorology, whereas our approach relies on backward trajectories. Since forward an backward trajectory approaches are known to sample cold temperature regions differently it is likely that such differences also cause related differences in reconstructed water vapor mixing ratios. Regarding the Konopka et al. (2023) study, the use of microphysics in box model calculations was included, but it was applied to a much smaller ensemble of data based on a single Geophysical flight during the StratoClim campaign. In their study, they compared forward calculations, initiated in air masses where both water vapor and ice were observed, with the nearest MLS and CALIPSO observations along such forward trajectories. Their results showed a negligible importance of ice, with the best results obtained when simple LDPs along the backward trajectories, starting from the observation point, were calculated. Full microphysical studies using CLaMS-Ice are numerically expensive and require additional assumptions related to ice (noting that observations of ice were available for the StratoClim campaign). We have now incorporated a new discussion of these aspects into the first paragraph in Section 4, explicitly addressing the role of supersaturation, trajectory sampling effects, and missing mixing processes in influencing the reconstruction bias.

Comment 2: One very interesting result is presented in Fig. 5. It shows that while the vast majority of the Lagrangian cold points upstream of the observations in the ASM are within the ASM, the NAM is a very different story. Although a small fraction of the NAM LCPs come from the NAM region, the majority of the LCPs lie within the ASM. This is an important finding since it emphasizes the dominant role of the Asian monsoon in controlling the moisture

entering the stratosphere in boreal summer, while the North American monsoon is relatively speaking a bit player. This result could well be highlighted more explicitly in the Conclusions section.

Thank you for this valuable suggestion. We agree that this finding is significant and have revised both the Abstract and the Discussion section to better highlight it. However, given that the reconstruction does not perform well over the NAM region, it could well be that the inferred long-range transport may be overestimated, potentially contributing to the limited reconstruction performance there. We have emphasized these results more clearly and included a related balanced discussion in the revised manuscript.

Comment 3: In Section 4, the authors address the finding of Randel et al. (2015). They are able to repeat the Randel et al. results with the satellite water vapor observations but not with the reconstructed water vapor fields. They argue that the simple Lagrangian approach fails to properly capture the effects of convection and ice injection in monsoon regions. This is not a convincing argument given that Konopka et al. (2023) did not find that convective processes played a significant role in determining the final stratospheric water vapor entry values in the circulation around the "dehydration carousel" in the Asian summer monsoon anticyclone.

Thank you for your comment. Based on our results in Fig. 1 and the newly added bars in Fig. 2, which present the anomalies of both observations and reconstructions, we now provide a clearer comparison. Additionally, we have included the heights of the cold point tropopause and lapse rate tropopause derived from ERA5 (the cyan and yellow lines in Fig. 2) for better understanding. Our analysis shows that the reconstruction performs well from about 16.5km upwards, capturing nearly the full magnitude of the anomalies in the ASM despite the systematic dry bias. This finding aligns with Konopka et al. (2023), who demonstrated that convection plays a limited role in setting final stratospheric water vapor concentrations within the Asian summer monsoon anticyclone. However, our reconstruction is less accurate below the tropopause. To enhance clarity, we have also added tropopause heights to Fig. 7, similar to Fig. 2, which makes it evident that below 16.5 km (in the upper troposphere, where the reconstruction method becomes less effective), convective processes significantly impact water vapor transport, and our method cannot capture this effect, explaining the discrepancies between reconstructed and observed water vapor. We acknowledge that this distinction was not emphasized clearly in our previous discussion. To address this, we have revised the second paragraph in Section 3.3 to explicitly state the limitations of our approach in capturing convective influences in particularly

the upper troposphere.

Comment 4: As a general comment, I found the narrative flow of the text choppy and confusing, particularly in the Introduction. The Introduction certainly recognizes the long-standing consensus that the dominant control on the concentration of water vapor entering is through slow horizontal transport. However, this is restated in various ways multiple times, suggesting a controversy that does not exist. At minimum, I would recommend a revision of the Introduction to make it shorter and read more smoothly.

Thank you for your feedback. We have removed the overly long introduction and comparison of the Lagrangian and local methods and have adjusted the narrative sequence to enhance readability.

**Minor Comments**

Comment 1: I found the discussion of the methodology of the water vapor reconstructions (Section 2.3.3) incomplete. They have adopted three types of reconstructions ("experiments"): two obtaining water vapor values from the CPT along back trajectories ("LAG_single" and "LAG") and a third based upon local CPTs ("LOC"). The first two types of reconstructions appear to be similar in approach to the Lagrangian trajectories used in similar studies going back at least two decades. However, the method by which the "LOC" reconstructions are carried out is unclear, especially since it lumps all three of the reconstruction approaches into one paragraph.

We appreciate this comment and have revised Section 2.3.3 to clarify the methodology regarding the LOC reconstruction.

Comment 2: It would have been helpful if the captions for Figs. 1 & 2 specifically stated that the reconstructions were obtained through the LAG "experiment."

Thank you for this suggestion. We have updated the captions for Figures 1 and 2.

Comment 3: (line 31) Pan et al. (2018) did not introduce the Lagrangian Cold Point, although they do provide a number of references to the Lagrangian approach to determining the effective dehydration temperature for air parcels entering the stratosphere.

We appreciate this correction and have revised the text.

Comment 4: (L.25–36) Taken together, these sentences comprising the latter half of the first paragraph in the Introduction restate the importance of the concept of the transport history of air parcels entering the stratosphere several times over. The paragraph gives little sense of what new insights the Lagrangian perspective has provided since earlier papers.

Thank you for highlighting this issue. We have revised the paragraph to streamline the discussion and focus on the new insights provided by the Lagrangian perspective.

**References**

Konopka, P., Rolf, C., Von Hobe, M., Khaykin, S. M., Clouser, B., Moyer, E., Ravegnani, F., D'Amato, F., Viciani, S., Spelten, N., Afchine, A., Krämer, M., Stroh, F., and Ploeger, F. (2023). The dehydration carousel of stratospheric water vapor in the asian summer monsoon anticyclone. *Atmospheric Chemistry and Physics*, 23(20):12935 – 12947.

Ploeger, F., Fueglistaler, S., Grooß, J.-U., Günther, G., Konopka, P., Liu, Y., Uller, R., Ravegnani, F., Schiller, C., Ulanovski, A., and Riese, M. (2011). Insight from ozone and water vapour on transport in the tropical tropopause layer (ttl). *Atmospheric Chemistry and Physics*, 11(1):407–419. cited By 54.

Poshyvailo, L., Müller, R., Konopka, P., Günther, G., Riese, M., Podglajen, A., and Ploeger, F. (2018). Sensitivities of modelled water vapour in the lower stratosphere: Temperature uncertainty, effects of horizontal transport and small-scale mixing. *Atmospheric Chemistry and Physics*, 18(12):8505–8527. cited By 18.

Schiller, C., Groob, J.-U., Konopka, P., Plöger, F., Silva Dos Santos, F., and Spelten, N. (2009). Hydration and dehydration at the tropical tropopause. *Atmospheric Chemistry and Physics*, 9(24):9647–9660. cited By 73.

Schoeberl, M. R., Dessler, A. E., and Wang, T. (2013). Modeling upper tropospheric and lower stratospheric water vapor anomalies. *Atmospheric Chemistry and Physics*, 13(15):7783–7793.

---

## Referee Report (RR1)

The authors have satisfactorily addressed my previous comments. This work is a valuable contribution to the community's understanding of water vapor anomalies above Northern Hemisphere summer monsoons. I have a few minor comments that the authors should address prior to publication.

~Stephen Bourguet

**Minor comments:**

- -Line 9: "water vapor is predominantly controlled by local temperatures near the tropopause in the Asian Monsoon." Does this refer to water vapor across the tropics, or just water vapor near the Asian Monsoon?
- -Line 14: "an underestimation of moistening due to convective ice injection may play a role in this region." Is there a reason to suggest that the link between convection and the reconstruction dry bias is specifically driven by ice injection? I would suggest making this statement more general if not.
- -Line 43: "Our goal is to evaluate the role of the freeze-drying mechanism in the large-scale temperature and wind fields for the enhancement of stratospheric water vapor over the ASM and NAM regions from a Lagrangian perspective." This study explores water vapor anomalies throughout the UTLS, not just the stratosphere. I think the region of the study (i.e., the UTLS) should be made clear here to prevent readers from focusing only on results in the stratosphere.
- -Line 55 and throughout the text: The term "deep tropics" is misused. This refers to the region within the tropics closest to the equator (e.g., 10S to 10N). It would be sufficient to just refer to the study region (35S to 35N) as the tropics.
- -Line 65: "How well can stratospheric water vapor mixing ratios in the ASM and NAM as observed by SAGE III/ISS and MLS be reconstructed using a simplified Lagrangian modelling method...." Similar to a previous comment does this study aim to focus on the stratosphere or the UTLS? This distinction matters for the analysis, especially given the emphasis on reconstructions at 16.5 km in Figures 1, 5, 6, and 8. (16.5 km is below the cold point tropopause across the tropics.) If the study is intended to focus on the stratosphere, then the analysis needs to be done above the cold point tropopause.
- -Line 205: The term "tropical tropopause layer" is used incorrectly here and in the following discussion. The TTL spans 150 hPa to 70 hPa, or 14 km to 18.5 km. It is not the layer between the lapse rate and cold point tropopauses.
- -Line 236: "whereas small-scale mixing appears to be a more dominant contributor." This is not part of the analysis and is speculative. Why the focus on small-scale mixing here? What about, e.g., ice injection?
- -Lines 347–348: "The locations of LCPs for the tropics (Fig. 5a–b) resemble an ensemble of those found in the ASM and NAM, suggesting that dehydration predominantly occurs near the

monsoon regions." Figure 5 shows that the LCPs are concentrated around southeast Asia for each ensemble. It does not show LCPs for the tropical ensemble centered around the monsoons. Instead, Fig. 5 shows that cold trap dehydration mechanism (Holton and Gettelman, 2001) dominates the tropical trajectories (as is correctly noted in the following sentence).

- -Lines 365–366: "OLR  $\geq$  1.5 standard deviations" and "OLR  $\leq$  -1.5 standard deviations" would be more clear if stated as "OLR  $\geq$  1.5 standard deviations above the mean" and "OLR  $\geq$  1.5 standard deviations below the mean." (Assuming that the mean is used.)
- -Lines 438–442: This paragraph implies that the Lagrangian method is effective across the tropics. This needs to be qualified to acknowledge the method's ineffectiveness in capturing the NAM water vapor anomalies.
- -Line 465–466: "Hence, it is likely the underestimated moistening effect of ice injection of convection in the western region of the Asian monsoon which controls the dry bias of Lagrangian reconstructions in the ASM." Convective ice injection is not part of the analysis, so this statement is speculative. Why the focus on ice injection here? Is it known that convection over the western sector brings ice particles to the UTLS?

There are a handful of typographical errors that need to be addressed:

- -Line 8: "The main dehydration, region ..." should be "The main dehydration region, ..."
- -Line 13: "dry bias in reconstruction ..." would read better as "the dry bias in reconstructions ..."
- -Lines 59–60: Citations need to be in parentheses.
- -Line 260: comma before "thus" should be a semicolon.
- -Line 407: "simulates the conversion of excess water vapor to ice, and setting parcels to saturation within convection zones" should be "simulates the conversion of excess water vapor to ice by setting parcels to saturation within convection zones."

---

## Referee Report (RR2)

**Partial review of resubmission of egusphere-2024-3260:**

"Understanding Boreal Summer UTLS Water Vapor Variations: A Lagrangian perspective"

Hongyue Wang, Miejong Park, Mengchu Tao, Cristina Peña-Ortiz, Nuria Pilar Plaza, Felix

Ploeger and Paul Konopka

**Explanation for this partial review:**

This is not a complete review and does not therefore include a recommendation on its acceptability for publication. The paper does appear to present interesting results that are relevant to the topic of the controls on lower stratospheric water vapor. Unfortunately, the paper is not well-written, and as a result I found it was taking more time than I had available to do a fair assessment of the paper's scientific merit.

So, the following includes a brief summary of the paper and a detailed commentary on Sections 1 through 3.1.1 up through line 236.

**Summary:**

As in their original submission, the authors present results of Lagrangian back-trajectories calculated from ERA5 meteorological fields to reconstruct the horizontal and vertical distributions of water vapor in the tropical upper troposphere and lower stratosphere (UTLS) during boreal summer. The paper poses three questions. First, how well do their simplified Lagrangian back-trajectory method reproduce water vapor values observed by SAGE III/ISS and MLS in the Asian Summer Monsoon (ASM) and North American Monsoon (NAM)? Second, are these values locally controlled or represent freeze drying upstream? Third, is the general tendency to a dry bias in the reconstructions related to particular processes?

The reconstructions are carried out using the CLaMS trajectory module (Konopka et al., 2022). Back trajectories are calculated for 180 days using the CLaMS model's trajectory module and were initiated from the satellite measurement locations and times. Results are presented for comparisons of the reconstructions with SAGE-III and MLS water vapor values at the back-trajectory initiation points. The water vapor reconstructions are based upon coldpoint temperatures, identified either from the local vertical temperature profile or from the back-trajectory minimum temperature (the Lagrangian CPT). Three types of reconstructions are thus done based on the type of CPT: (a) using local CPTs (LOC), (b) using the Lagrangian CPT for every single trajectory (LAG\_single), and (c) using the average Lagrangian CPT for a cluster of 51 back trajectories.

The authors find that while both SAGE III and MLS reconstructions produce similar spatial patterns, the SAGE III reconstruction values tend to be higher than those for the MLS.

Reconstructions generally succeed in the ASM but they do not capture the NAM pattern. Reconstructions in the former region appear to be predominantly controlled locally while reconstructed water vapor in the NAM seems to be largely remotely controlled. Finally, the dry bias in the reconstructions over the ASM tends to increase with the intensity of convection.

**General comments:**

The revised submission addresses a topic that has received considerable attention over the past 20 years or so. And as in the original submission, it is a positive feature of the approach that the authors have undertaken in this work that they directly compare their water vapor reconstructions with SAGE-III ISS and MLS water vapor observations.

**Specific comments:**

**Section 1**

- **L.21:** Suggested rewording:
- "...where air masses undergo slow diabatic ascent ..."
- "...into the stratosphere over time scales of weeks to months."
- L36: What exactly is the tropical stratospheric water vapor anomaly? An anomaly with respect to what? Likewise for the summertime NH extratropical water vapor maximum.
- L41: Replace predict with assess.
- L42: The sentence here is confusing. The large-scale temperature and wind fields are no more than a *representation* of the atmosphere that the reanalysis provides. They do not have their own set of processes. The freeze-drying occurring in the actual atmosphere can of course be estimated with reanalysis wind and temperature fields that's the methodology here after all. But as I read the sentence, it implies that there is a freeze-drying mechanism for the large-scale fields and another at finer resolution.
- Lambert et al. (2017) is a nightmare of parentheses. Since this is only the introductory section, there is no immediate need to specify version numbers of datasets or their specific references. This can and should be the business of Section 2. In any case, by eliminating all those version numbers and references, the profusion of parentheses goes away. Furthermore, identifying the SAGE III and MLS versions here implies that the updates they represent were somehow critical to the outcome of the study, which I am pretty sure is not what you are saying.
- **L59:** This sentence amounts to an awkward juxtaposition inasmuch as you have just pointed out the advantage of the SAGE III data over MLS. The advantage of the MLS data of course is their unmatched temporal and spatial coverage. In this regard, SAGE III/ISS is a very poor cousin indeed.

**L60:** improper use of **furthermore.** Use another more appropriate conjunctive adverb.

**L68:** ...are most critical. Critical for what, exactly? Presumably remote control, but that is not stated.

**Section 2**

L83: It would be helpful here to describe the particular sampling challenge presented by the MLS data. A 10° latitude x 20° longitude grid box is a big piece of real estate but nonetheless at the latitudes of interest in this study, on most days it would be traversed by only a couple of MLS overpasses, one ascending (afternoon) and one descending (nighttime). The limb-viewing geometry also comes into play here, in particular the 200-km swath length along the orbit path and at the lower MLS levels (100 hPa and below) where spatial variability rears its ugly head. In short there are a number of temporal and spatial sampling considerations that almost certainly play a role in a three-dimensional gridding of MLS data, and the same can be said for SAGE III/ISS. Given their very different times at which they sample alone, MLS and SAGE III/ISS "see the world" as it were in different ways. Are these different perspectives significant? I don't know – I didn't do the work! But I think it's a fair question and should be addressed here.

**L88:** fix reference.

**L90:** missing terminal parenthesis, period and double space

**L97:** The latitude range 35°S-35°N is incorrectly referred to as the subtropics.

L97-L100: The previous subsection on MLS does not mention a specific period, so presumably the study period mentioned here, the months of August from 2017 to 2022, applies to both MLS and SAGE III/ISS. But perhaps not, as the second sentence in the paragraph says that data for the years 2020-2022 were added in order to get more spatial coverage. I find it hard to make sense of what's going on here unless there are actually two periods of study here: a 2017-2019 period for MLS and SAGE-III, and a supplemental 2020-2002 period just for SAGE III/ISS. If this is in fact the case, then the trustworthiness of the foregoing comparisons between the MLS and SAGE III/ISS reconstructions is seriously undermined. That it was the case that MLS data were not used in the latter period is strongly implied by the statement that there is no significant difference between the SAGE reconstruction results from the two periods.

This of course begs the question of why MLS data weren't used over the entire 2017-2022 period in the first place. If there is a technical reason for this, one isn't mentioned.

NOTE: The question of the two study periods is answered at **L131**. That this is referred to only obliquely here muddies the narrative unacceptably.

L115: In this paper, the term **anomaly** is used primarily for the difference between a derived value at a given geographical location from its long-term mean. However, here it refers instead to the difference between an (OLR) instantaneous value and its temporal mean. The latter usage is the more natural one as I see it, but in any case the dual usage here leads to some confusion in the text. In particular, back at L36 the phrase tropical stratospheric water vapor anomaly is in the same sentence as the summertime NH extratropical water vapor maximum. Is this maximum a spatial or temporal one? It's not clear.

I mention this since the spatial features in the water vapor fields that are associated with the ASM and NAM could easily be referred to as local maxima without any ambiguity.

- **L150:** Suggest replacing **along** with **in.** The word "along" is more suitable for horizontal spans or temporal stretches.
- **L151:** The SAGE III/ISS CPTs used in LOC are from the MERRA-2 reanalysis while the trajectory reconstructions use ERA5 reanalysis temperatures. This necessarily introduces a complication into any comparison between the LOC and two Lagrangian experiments.
- **L172:** Here is the first reference in the manuscript to the direction of crossing the tropopause. If it's not significant enough consideration to merit mention in the Introduction, why does it figure into the methodology?

**Section 3**

**Figure 1 (from L180):**

- The caption should identify the level of the water vapor fields in the panels. The reader should not need to refer to the text.
- The lower four panels are labeled as either **Observation anom** and **Reconstruction anom**. I would simply identify them as maps of the differences between the reconstructions and the satellite observations.
- The concluding clause in the caption is awkward English, the use of the word strings in particular.

¶1 (L181-L192) In keeping my preceding comment, the spatial features in the eight panels in Figure 1 are variously referred to as "enhanced", "elevated", "high", "maxima" and "anomalies". If, as I think is the case, all these terms essentially refer to the same thing, a single term should be used. Otherwise the text is ambigious.

**L195:** A simpler way to put this is that the reconstruction only faintly reproduces the observed NAM water vapor pattern.

**L196:** change **increase** to **elevation.** The word increase implies a temporal change.

Comment on the discussion of Figure 1: The spatial patterns displayed by the positive-valued reconstruction "anomalies" are a nice result, in particular their similarity to the maximum in the field of SAGE III observation in the vicinity ASM. While the text at L188 does note the overall dry bias of the reconstructions for each satellite dataset, one might expect that, outside of the regions like the ASM where the water vapor stratospheric entry values of water vapor are controlled locally, the reconstruction anomaly field would be more or less flat. Instead the negative-valued anomalies more or less mimic the patterns in the observations, in much the same way as the positive-valued anomalies.

I don't know if there is a simple explanation for this or not, but it seems whatever is causing coherence of the positive-values reconstruction anomalies and the observed fields may also be contributing to a coherence of the negative-values anomalies.

**Figure 2 (from L197):**

- (caption) replace concentrations with mixing ratio throughout.
- (caption) replace For each subplot, it shows with Each panel shows.
- (caption) Properly, biases are not "between" fields/variable, but the difference of a particular estimate of a variable/field from a "reference" or "true" value of the same.
- (caption) The phrase reconstructed values subtract observed values needs to be corrected. Suggest reconstructed minus observed.
- The lower panels are labeled as either **Observation anom** which suggests that identifying them as maps of the differences between the reconstructions and the satellite data would be preferable. Thus they hange **the portions of TST are shown with upper right strings of c-d and g-h** to **TST fractions noted in panels c-d and g-h**.

- **L202-206:** It is not surprising that the reconstructions do so poorly below the tropopause, as this where the remoistening by clouds will be a factor.
- **L221, L237:** What is the basis of the statement that in the ASM region the tropopause layer is higher and thinner than elsewhere in the tropics? Likewise that the NAM is also thinner but lower?
- **L224:** It's not clear what the phrase **vertical performance** means.
- **L225-L229:** As written, neither of these sentences make sense. What do **one-third of the observed anomalies** and **over two-thirds of the observed values** mean? I assume these mean the magnitudes of the reconstructions that are being referred to, but the phraseology is poorly chosen.
- L230:-L236: How is the "consistent behavior of the reconstruction in the ASM compared to the tropics" evidence for the water vapor above the ASM by something called mechanisms-freeze-drying in the large-scale temperature. This is its first mention in the paper, and if it is indeed the 'advection-condensation' paradigm of Liu et al. (2010), is there a reason it needs to be given a new name and a grammatically unpleasant one at that? In any case, what exactly is this mechanisms-freeze-drying in the large-scale temperature process and how it is related to the "consistent behavior" above?

---

## Author Response (AR2)

**Response to Referee 1**

We thank the reviewer for the careful reading of our manuscript and the constructive feed-back. In revising the paper, we went through the entire manuscript to improve the wording, consistency, and overall clarity. Below, we address each of the reviewer's comments in detail, indicating the changes made in the revised version.

**General Comments**

The authors have satisfactorily addressed my previous comments. This work is a valuable contribution to the community's understanding of water vapor anomalies above Northern Hemisphere summer monsoons. I have a few minor comments that the authors should address prior to publication.

**Minor comments**

• Line 9: "water vapor is predominantly controlled by local temperatures near the tropopause in the Asian Monsoon." Does this refer to water vapor across the tropics, or just water vapor near the Asian Monsoon?

Thank you for pointing this out. We have changed the sentence to "... water vapor in the Asian Monsoon is predominantly controlled by local tropopause temperatures" (L8-9).

• Line 14: "an underestimation of moistening due to convective ice injection may play a role in this region." Is there a reason to suggest that the link between convection and the reconstruction dry bias is specifically driven by ice injection? I would suggest making this statement more general if not.

We removed the specific "ice injection" (L14).

• Line 43: "Our goal is to evaluate the role of the freeze-drying mechanism in the large-scale temperature and wind fields for the enhancement of stratospheric water vapor over the ASM and NAM regions from a Lagrangian perspective." This study explores water vapor anomalies

throughout the UTLS, not just the stratosphere. I think the region of the study (i.e., the UTLS) should be made clear here to prevent readers from focusing only on results in the stratosphere. We have changed the stratospheric water vapor to UTLS water vapor for this sentence (L42-43), and we also modified other texts that have similar issue.

• Line 55 and throughout the text: The term "deep tropics" is misused. This refers to the region within the tropics closest to the equator (e.g., 10S to 10N). It would be sufficient to just refer to the study region (35S to 35N) as the tropics.

Thanks for clarifying this. We have modified the corresponding texts at L44.

• Line 65: "How well can stratospheric water vapor mixing ratios in the ASM and NAM as observed by SAGE III/ISS and MLS be reconstructed using a simplified Lagrangian modelling method..." Similar to a previous comment — does this study aim to focus on the stratosphere or the UTLS? This distinction matters for the analysis, especially given the emphasis on reconstructions at 16.5 km in Figures 1, 5, 6, and 8. (16.5 km is below the cold point tropopause across the tropics.) If the study is intended to focus on the stratosphere, then the analysis needs to be done above the cold point tropopause.

Our focus is on the UTLS region. We thank the reviewer for noting the confusing formulation and have revised the first research question to read: "How well can UTLS water vapor mixing ratios in the ASM and NAM be reconstructed ..." (L44–45).

• Line 205: The term "tropical tropopause layer" is used incorrectly here and in the following discussion. The TTL spans 150 hPa to 70 hPa, or 14 km to 18.5 km. It is not the layer between the lapse rate and cold point tropopauses.

We avoided using this term wrongly (L178-179).

• Line 236: "whereas small-scale mixing appears to be a more dominant contributor." This is not part of the analysis and is speculative. Why the focus on small-scale mixing here? What about, e.g., ice injection?

Sorry for the confusion. Here, we mentioned that the small-scale mixing was based on Plaza

et al. (2021). To improve clarity and flow, we have removed this sentence in the revised manuscript (L189-191).

• Lines 347–348: "The locations of LCPs for the tropics (Fig. 5a–b) resemble an ensemble of those found in the ASM and NAM, suggesting that dehydration predominantly occurs near the monsoon regions." Figure 5 shows that the LCPs are concentrated around southeast Asia for each ensemble. It does not show LCPs for the tropical ensemble centered around the monsoons. Instead, Fig. 5 shows that cold trap dehydration mechanism (Holton and Gettelman, 2001) dominates the tropical trajectories (as is correctly noted in the following sentence).

We agree with this and have revised the text to read: "For the ASM (Fig. ??c), the LCPs

cluster near the monsoon region and exhibit high reconstructed mixing ratios consistent with elevated cold-point temperatures. For the NAM (Fig. ??e), although some LCPs are located in its vicinity, a considerable portion is concentrated over southern Asia." (L236–238)

- Lines 365–366: "OLR  $\geq$  1.5 standard deviations" and "OLR  $\leq$  -1.5 standard deviations" would be more clear if stated as "OLR  $\geq$  1.5 standard deviations above the mean" and "OLR  $\leq$  1.5 standard deviations below the mean." (Assuming that the mean is used.) Modified (L261-262).
- Lines 438–442: This paragraph implies that the Lagrangian method is effective across the tropics. This needs to be qualified to acknowledge the method's ineffectiveness in capturing the NAM water vapor anomalies.

Sorry that we didn't express the information clearly enough. Now the paragraph has been reorganized (L320-322).

• Line 465–466: "Hence, it is likely the underestimated moistening effect of ice injection of convection in the western region of the Asian monsoon which controls the dry bias of Lagrangian reconstructions in the ASM." Convective ice injection is not part of the analysis, so this statement is speculative. Why the focus on ice injection here? Is it known that convection over the western sector brings ice particles to the UTLS?

We avoid using this speculative statement. Now the sentence is modified to read: "These findings suggest that underestimated moistening from convection in the western part of the ASM is a key driver of the dry bias in the reconstructions." (L337-338)

- There are a handful of typographical errors that need to be addressed:
  - Line 8: "The main dehydration, region ..." should be "The main dehydration region, ..."
  - Line 13: "dry bias in reconstruction ..." would read better as "the dry bias in reconstructions ..."
  - Lines 59–60: Citations need to be in parentheses.
  - Line 260: comma before "thus" should be a semicolon.
  - Line 407: "simulates the conversion of excess water vapor to ice, and setting parcels to saturation within convection zones" should be "simulates the conversion of excess water vapor to ice by setting parcels to saturation within convection zones."

We thank the reviewer for identifying these typographical issues. They have been corrected accordingly.

**Response to Referee 3**

We thank the reviewer for the careful reading and the thoughtful and detailed feedback. We agree that some of the wording and formulations may have caused unnecessary confusion and could still be improved. Following the overall recommendation of the reviewer, we thoroughly worked through the manuscript text again to enhance clarity and conciseness. A few examples of changes which apply to the overall manuscript are summarized here:

- In the revised version of the manuscript, we follow the recommendation of the reviewer and consistently use the term "advection-condensation paradigm" to describe the methodology applied in this study. In this sense, we avoid the usage of different terms to describe the method, as was done before (e.g., freeze-drying, advection-condensation, etc.).
- Similarly, the use of the term "anomaly,", which may have been unclear before, has been changed. In the revised version, we clearly define the type of anomaly before using the term, especially distinguishing between spatial and temporal anomalies, as this distinction is essential for interpreting the results correctly.
- Overall, we tried to shorten sentences, removed parentheses, etc, to simplify the wording.

More detailed replies to the specific points are given further below. We emphasize, that we not only considered these specific points provided by the reviewer but worked over the entire manuscript to improve the text consistently. We are confident that these changes significantly improved the manuscript such that it is ready for publication now.

**Specific comments**

- L21: Suggested rewording: "... where air masses undergo slow diabatic ascent ..." "... into the stratosphere over timescales of weeks to months."

  Revised (L20-21).
- L36: What exactly is the tropical stratospheric water vapor anomaly? An anomaly with respect to what? Likewise for the summertime NH extratropical water vapor maximum.

We appreciate the reviewer's comment and acknowledge that the wording was imprecise. The ASM contribution to the tropical stratospheric tape-recorder moist signal has been quantified as 25% by Bannister et al. (2004) and 14% by Nützel et al. (2019). However, since we have restructured the Introduction, this discussion has now been removed.

• L41: Replace *predict* with *assess*.

Revised (L34).

• L42: The sentence here is confusing. The large-scale temperature and wind fields are no more than a representation of the atmosphere that the reanalysis provides. They do not have their own set of processes. The freeze-drying occurring in the actual atmosphere can of course be estimated with reanalysis wind and temperature fields—that's the methodology here after all. But as I read the sentence, it implies that there is a freeze-drying mechanism for the large-scale fields and another at finer resolution.

We thank the reviewer for pointing this out. To avoid confusion, we have revised the sentence to read: "... we aim at evaluating dehydration processes in the UTLS over the ASM and NAM regions from a Lagrangian perspective." (L42-43)

• L50: The phrase beginning with *to reconstruct* ... and ending with ... *Version 5.0 Lambert et al.* (2017) is a nightmare of parentheses. Since this is only the introductory section, there is no immediate need to specify version numbers of datasets or their specific references. This can and should be the business of Section 2. In any case, by eliminating all those version numbers and references, the profusion of parentheses goes away. Furthermore, identifying the SAGE III and MLS versions here implies that the updates they represent were somehow critical to the outcome of the study, which I am pretty sure is not what you are saying.

Thank you the specific suggestion. We have revised the text accordingly (L49-55).

• L59: This sentence amounts to an awkward juxtaposition, inasmuch as you have just pointed out the advantage of the SAGE III data over MLS. The advantage of the MLS data, of course, is their unmatched temporal and spatial coverage. In this regard, SAGE III/ISS is a very poor

cousin indeed.

We agree that MLS provides much greater temporal and spatial coverage. SAGE III/ISS offers finer vertical resolution (about 2 km, due to the 1–2–1 smoothing, L92-94), which permits more vertical structure to be depicted on a 0.5 km grid. To remove the juxtaposition, we have revised the text to state the MLS coverage advantage and the SAGE III/ISS vertical-resolution separately, without suggesting equivalence (L50-52).

- L60: Improper use of *furthermore*. Use another, more appropriate, conjunctive adverb. We have re-organized the paragraph and now use "Finally" as a short transition (L57).
- L68: ... are most critical. Critical for what, exactly? Presumably remote control, but that is not stated.

Sorry for causing confusion. The sentence has been changed to: "Are the moisture enhancements observed within the ASM and NAM anticyclones locally or remotely controlled and which regions contribute most strongly to these enhancements?" (L46-47)

• L83: It would be helpful here to describe the particular sampling challenge presented by the MLS data. A 10° latitude × 20° longitude grid box is a big piece of real estate, but nonetheless at the latitudes of interest in this study, on most days it would be traversed by only a couple of MLS overpasses, one ascending (afternoon) and one descending (nighttime). The limb-viewing geometry also comes into play here, in particular the 200 km swath length along the orbit path and at the lower MLS levels (100 hPa and below) where spatial variability rears its ugly head. In short, there are a number of temporal and spatial sampling considerations that almost certainly play a role in a three-dimensional gridding of MLS data, and the same can be said for SAGE III/ISS. Given their very different times at which they sample alone, MLS and SAGE III/ISS "see the world," as it were, in different ways. Are these different perspectives significant? I don't know—I didn't do the work! But I think it's a fair question and should be addressed here.

We thank the reviewer for the detailed comment. We acknowledge that MLS and SAGE III/ISS differ in sampling characteristics, including spatial coverage and observation times, and we appreciate the concern. To clarify, we have revised the text (L71–72, L94–95), that horizontal

gridding was applied only in Fig. 1 to illustrate the horizontal distributions. For the vertical profiles in Fig. 2 and Fig. 7, data were averaged within the defined horizontal boxes. For all other analyses, no horizontal averaging was performed. In particular, for the CLaMS trajectory simulations, air parcels were released at the exact observing location and time of each profile, and correlation coefficients were calculated using individual points. Therefore, despite their different sampling strategies, we consider the reconstruction performance of MLS and SAGE III/ISS to be comparable. For experiment LAG, we added 50 more trajectories around each observation point to enlarge the ensembles vertically, covering  $H\pm0.25\,\mathrm{km}$  (H is the height of the observation point) for both SAGE III/ISS and MLS. We recognize, however, a remaining incomparability, as we did not account for the large vertical kernel used for MLS. We will keep this in mind and consider a better approach in future work.

• L88: Fix reference.

Corrected (L75).

• L90: Missing terminal parenthesis, period, and double space.

Corrected (L77).

• L97: The latitude range 35°S–35°N is incorrectly referred to as the subtropics.

We corrected the "subtropics" to "tropics" (L84).

• L97-L100: The previous subsection on MLS does not mention a specific period, so presumably the study period mentioned here, the months of August from 2017 to 2022, applies to both MLS and SAGE III/ISS. But perhaps not, as the second sentence in the paragraph says that data for the years 2020 -2022 were added in order to get more spatial coverage. I find it hard to make sense of what's going on here unless there are actually two periods of study here: a 2017 -2019 period for MLS and SAGE -III, and a suppl emental 2020-2002 period just for SAGE III/ISS. If this is in fact the case, then the trustworthiness of the foregoing comparisons between the MLS and SAGE III/ISS reconstructions is seriously undermined. That it was the case that MLS data were not used in the latter period is strongly implied by the statement that there is

no significant difference between the SAGE reconstruction results from the two periods. This of course begs the question of why MLS data weren't used over the entire 2017-2022 period in the first place. If there is a technical reason for this, one isn't mentioned.

NOTE: The question of the two study periods is answered at L131. That this is referred to only obliquely here muddies the narrative unacceptably.

We thank the reviewer for this helpful comment. We agree that the distinction between the study periods was not sufficiently clear in the original version. In the revised manuscript, we now explicitly state at the beginning of the subsection that MLS data are analyzed for 2017–2019, while SAGE III/ISS data are extended to 2017–2022 to ensure sufficient spatial coverage (L84-86). We also clarify the technical reason for using the shorter MLS period: the large volume of MLS data makes it computationally prohibitive to perform all trajectory calculations over a longer time span, particularly given that we launch large ensembles for each measurement point (the specific trajectory numbers are provided in Section 2.3.1). To address the reviewer's concern regarding the robustness of the comparison, we now include a direct comparison of SAGE III/ISS reconstructions using only 2017–2019 versus the full 2017–2022 period (Fig. S1). The results show no significant differences, demonstrating that the extended SAGE III/ISS period does not affect our conclusions and thus justifies our choice to continue with the 2017–2022 period for SAGE III/ISS.

• L115: In this paper, the term anomaly is used primarily for the difference between a derived value at a given geographical location from its long-term mean. However, here it refers instead to the difference between an (OLR) instantaneous value and its temporal mean. The latter usage is the more natural one as I see it, but in any case the dual usage here leads to some confusion in the text. In particular, back at L36 the phrase tropical stratospheric water vapor anomaly is in the same sentence as the summertime NH extratropical water vapor maximum. Is this maximum a spatial or temporal one? It's not clear. I mention this since the spatial features in the water vapor fields that are associated with the ASM and NAM could easily be referred to as local maxima without any ambiguity.

We thank the reviewer for pointing this out. We now explicitly state in the manuscript whether a anomaly or maxima is spatial or temporal (e.g., at L102–104 we specify temporal OLR anomalies). The issue raised at L36 is addressed in our earlier response.

• L150: Suggest replacing along with in. The word "along" is more suitable for horizontal spans or temporal stretches.

Revised (L136).

• L151: The SAGE III/ISS CPTs used in LOC are from the MERRA-2 reanalysis while the trajectory reconstructions use ERA5 reanalysis temperatures. This necessarily introduces a complication into any comparison between the LOC and two Lagrangian experiments.

We thank the reviewer for pointing this out. In the original version, we used MERRA-2 temperature profiles because SAGE III/ISS water vapor data are reported as number densities on altitude and converted to mixing ratios using MERRA-2 temperature and pressure profiles (Park et al., 2021). We acknowledge that this introduced an inconsistency with the ERA5-based trajectory reconstructions. Upon checking, we found that CPTs from MERRA-2 and ERA5 are highly consistent; nevertheless, to avoid unnecessary complexity we now use ERA5 temperatures throughout and have updated the corresponding text (L136) and figures (Fig. 3 and Fig. 4) accordingly.

• L172: Here is the first reference in the manuscript to the direction of crossing the tropopause. If it's not significant enough consideration to merit mention in the Introduction, why does it figure into the methodology?

The explanation for TST in the methonds section is mainly to distinguish air parcels that are transported from the troposphere into the stratosphere from those circulating within the stratosphere. Since our results used both TST and non-TST trajectories and they are reconstructed differently, we consider this needs to be explicitly described in the methods. In the Introduction, we have now stated the direction of tropopause crossing when introducing the Lagrangian perspective (L36-38).

**• Figure 1 (from L180):**

- The caption should identify the level of the water vapor fields in the panels. The reader should not need to refer to the text.
- The lower four panels are labeled as either Observation anom and Reconstruction anom.

I would simply identify them as maps of the differences between the reconstructions and the satellite observations.

- The concluding clause in the caption is awkward English, the use of the word strings in particular.
- The lower panels are labeled as "Observation anom," which suggests that identifying them as maps of the differences between the reconstructions and the satellite data would be preferable. Thus change the portions of TST shown with upper right strings of c–d and g–h to "TST fractions noted in panels c–d and g–h."

We thank the reviewer for the suggestions. The anomalies in panels (e–h) are spatial anomalies relative to the tropical mean of the corresponding distributions in panels (a–d). Instead of using the difference between the reconstructions and observations, we use the anomalies to highlight that the reconstruction can reproduce the anomalies in the ASM, though with a commen dry bias. If using the difference, the dry bias will diminish the presentation of the good performance of the reconstruction.

We now have revised the caption to: Horizontal distributions of water vapor concentrations and anomalies in August. Panels (a–b) show observed H2O concentrations, (c–d) reconstructed concentrations from Experiment LAG, and (e–h) the corresponding spatial anomalies, based on SAGE III/ISS at 16.5 km (left) and MLS at ~16.3 km (right). The anomalies are calculated relative to the tropical mean (35°S–35°N)...

• L181-L192: In keeping with my preceding comment, the spatial features in the eight panels in Figure 1 are variously referred to as "enhanced", "elevated", "high", "maxima" and "anomalies". If, as I think is the case, all these terms essentially refer to the same thing, a single term should be used. Otherwise the text is ambiguous.

We thank the reviewer for noting this inconsistency. In the revised manuscript, we use the terminology more consistently: "maxima" to describe the high values over the monsoon regions in Fig. 1a–d, and "moisture enhancement" for the positive anomalies in Fig. 1e–h (L165–173).

• L195: A simpler way to put this is that the reconstruction only faintly reproduces the observed NAM water vapor pattern.

Revised (L170).

• L196: Change increase to elevation. The word increase implies a temporal change.

We modified the sentence according to the suggestion above and thus avoided using the word "increase." (L170)

• Comment on the discussion of Figure 1: The spatial patterns displayed by the positive-valued reconstruction "anomalies" are a nice result, in particular their similarity to the maximum in the field of SAGE III observation in the vicinity of the ASM. While the text at L188 does note the overall dry bias of the reconstructions for each satellite dataset, one might expect that outside regions like the ASM, where the stratospheric entry values of water vapor are controlled locally, the reconstruction anomaly field would be more or less flat. Instead, the negative-valued anomalies more or less mimic the patterns in the observations, in much the same way as the positive-valued anomalies. I do not know if there is a simple explanation for this or not, but it seems that whatever is causing the coherence of the positive-valued reconstruction anomalies with the observed fields may also be contributing to a coherence of the negative-valued anomalies.

We are very pleased that the Lagrangian reconstruction captures the positive anomalies well, especially within the ASM region. This can be explained by temperature-driven condensation processes that are well represented in the ERA5 temperature fields. As pointed out by the reviewer, this also appears to be the case for the negative anomalies. The explanation is essentially the same: the reconstruction reflects the temperature history along the trajectories. Even though the backward trajectories are generally longer for this more stratospheric part of the domain, the coherence with the observed patterns remains. In summary, this highlights the strength of the Lagrangian reconstruction—already noted in many earlier studies—in its ability to reproduce stratospheric water vapor anomalies based on the Lagrangian temperature history.

- Figure 2 (from L197):
  - (caption) Replace concentrations with mixing ratio throughout.

- (caption) Replace "For each subplot, it shows" with "Each panel shows."
- (caption) Properly, biases are not "between" fields/variables, but the difference of a particular estimate of a variable/field from a "reference" or "true" value of the same.
- (caption) The phrase "reconstructed values subtract observed values" needs to be corrected. Suggest "reconstructed minus observed."

We have applied the wording changes as suggested in the revised manuscript.

• L202–206: It is not surprising that the reconstructions do so poorly below the tropopause, as this is where remoistening by clouds is a factor.

We agree. Now we have added a short sentence to discuss this (L176-178).

• L221, L237: What is the basis of the statement that in the ASM region the tropopause layer is higher and thinner than elsewhere in the tropics? Likewise, that the NAM is also thinner but lower?

Sorry for causing the confusion. We meant the 'tropopause layer' as the layer between the lapse rate tropopause and the cold point tropopause, which are indicated by the yellow and cyan lines. Now we consider such description was not appropriate and helpful, so we have removed them (L182, L192).

• L224: It is not clear what the phrase vertical performance means.

We rephrased the sentence to read: "The contrast between the coral and grey bars in the right sub-panels illustrates how well the moisture enhancements relative to the tropical mean are captured." (L183–184)

• L225–L229: As written, neither of these sentences makes sense. What do "one-third of the observed anomalies" and "over two-thirds of the observed values" mean? I assume these refer to the magnitudes of the reconstructions, but the phrasing is poorly chosen.

We have revised this to read:" At 15.5 km, reconstructions based on SAGE III/ISS reproduce about one-third of the observed enhancement magnitude. Agreement improves with altitude,

exceeding two-thirds at 16.5 km and approaching close consistency above this altitude." (L184-186)

• L230–L236: How is the "consistent behavior of the reconstruction in the ASM compared to the tropics" evidence for the water vapor above the ASM being affected by something called "mechanisms-freeze-drying in the large-scale temperature"? This is its first mention in the paper, and if it is indeed the "advection–condensation" paradigm of Liu et al. (2010), is there a reason it needs to be given a new name—and a grammatically unpleasant one at that? In any case, what exactly is this "mechanisms-freeze-drying in the large-scale temperature" process and how is it related to the "consistent behavior" noted above?

We thank the reviewer for pointing out this issue. The phrase "mechanisms–freeze-drying in the large-scale temperature" was an inappropriate wording. In the revised manuscript, we first introduce the advection–condensation paradigm at L35 and consistently refer to this established term throughout (e.g., at L187–188).

**References**

Park, M., Randel, W. J., Damadeo, R. P., Flittner, D. E., Davis, S. M., Rosenlof, K. H., Livesey, N., Lambert, A., and Read, W. (2021). Near-global variability of stratospheric water vapor observed by sage iii/iss. *Journal of Geophysical Research: Atmospheres*, 126(7). Cited by: 6; All Open Access, Green Open Access.